# Organogermanium THGP Induces Differentiation into M1 Macrophages and Suppresses the Proliferation of Melanoma Cells via Phagocytosis

**DOI:** 10.3390/ijms24031885

**Published:** 2023-01-18

**Authors:** Junya Azumi, Tomoya Takeda, Yasuhiro Shimada, Tao Zhuang, Yoshihiko Tokuji, Naoya Sakamoto, Hisashi Aso, Takashi Nakamura

**Affiliations:** 1Research Division, Asai Germanium Research Institute Co., Ltd., Suzuranoka 3-131, Hakodate 042-0958, Japan; 2Laboratory of Animal Health Science, Graduate School of Agricultural Science, Tohoku University, 468-1 Aoba, Aramaki, Aoba-ku, Sendai 980-8572, Japan; 3Department of Human Sciences, Obihiro University of Agriculture and Veterinary Medicine, Nishi 2 Sen, Inada, Obihiro 080-8555, Japan; 4Isotope Imaging Laboratory, Creative Research Institution, Hokkaido University, Kita 10 Jo-Nishi 5, Kita, Sapporo 060-0810, Japan

**Keywords:** organogermanium, Ge-132, THGP, M1 macrophage, melanoma

## Abstract

M1 macrophages are an important cell type related to tumor immunology and are known to phagocytose cancer cells. In previous studies, the organogermanium compound poly-*trans*-[(2-carboxyethyl)germasesquioxane] (Ge-132) and its hydrolysate, 3-(trihydroxygermyl) propanoic acid (THGP), have been reported to exert antitumor effects by activating NK cells and macrophages through the induction of IFN-γ activity in vivo. However, the detailed molecular mechanism has not been clarified. In this study, we found that macrophages differentiate into the M1 phenotype via NF-κB activation under long-term culture in the presence of THGP in vitro and in vivo. Furthermore, long-term culture with THGP increases the ability of RAW 264.7 cells to suppress B16 4A5 melanoma cell proliferation. These mechanisms indicate that THGP promotes the M1 polarization of macrophages and suppresses the expression of signal-regulatory protein alpha (SIRP-α) in macrophages and CD47 in cancers. Based on these results, THGP may be considered a new regulatory reagent that suppresses tumor immunity.

## 1. Introduction

Macrophages are classified as M1-type (proinflammatory) or M2-type (anti-inflammatory) macrophages [1,2]. Nonactivated macrophages (M0 macrophages or naïve macrophages) differentiate in response to IFN-γ secreted by CD4^+^ helper T cells (Th1 T cells), CD8^+^ T cells (cytotoxic T cells) and natural killer cells [3]. M1 macrophages show a strong ability to present antigens and phagocytose foreign matter, and they contribute to the secretion of inflammatory cytokines, such as tumor necrosis factor alpha (TNF-α), interleukin-1 (IL-1) and interleukin-6 (IL-6), resulting in the removal of bacteria and viruses [4]. In contrast, M2 macrophages differentiate in response to interleukin-4 (IL-4) secreted by Th2 cells and regulatory T cells (Tregs). M2 macrophages play roles in wound healing and immune tolerance through the secretion of transforming growth factor-β (TGF-β) and interleukin-10 (IL-10). M1/M2 macrophages are in a constant balance, and when this balance is lost, various inflammatory diseases develop [5].

Macrophages also play an important role in cancer [6]. M1 macrophages enhance immunity and phagocytosis by targeting tumors [7,8]. Moreover, M1 macrophages suppress cancer progression and metastasis [9]. Therefore, research on M1 macrophages has been performed in the field of cancer therapy [10]. However, cancer cells express CD47 to avoid recognition by macrophages. CD47 is known as a “don’t eat me signal” and binds to signal-regulatory protein alpha (SIRP-α) expressed on macrophages, allowing cancer cells to escape phagocytosis by macrophages [11]. CD47 is expressed at high levels in various types of cancer cells compared to normal cells [12,13,14]. Furthermore, inhibitory antibodies against CD47 or SIRP-α increase phagocytosis by macrophages and suppress cancer progression [15]. Conversely, M2-like macrophages, which are known as tumor-associated macrophages (TAMs), possess protumorigenic activity [16]. Cancer cells promote the differentiation of macrophages into TAMs to generate the tumor microenvironment. In particular, TAMs promote cancer metastasis by secreting Activin A, which is a member of the TGF family, and increase angiogenesis by secreting vascular endothelial growth factor (VEGF) [17]. Therefore, the suppression of TAMs is an important target in cancer therapy [18].

Melanoma is a malignant tumor with a low morbidity rate, but it is a very fatal disease [19,20,21]. In addition to the effectiveness of molecular-targeted drugs targeting BRAF in melanoma cells, immune checkpoint inhibitors targeting PD-1/PD-L1 have also been used [22]. Furthermore, immunotherapy inhibiting the CD47–SIRP-α interaction has also been reported to be effective against melanoma [23].

The organogermanium compound poly-*trans*-[(2-(carboxyethyl)germasesquioxane] (Ge-132; repagermanium) and its hydrolysate 3-(trihydroxygermyl)propanoic acid (THGP) are reported to possess various physiological activities [24]. In addition, Ge-132 exerts antitumor effects on both mice and rats in vivo [25,26,27]. The antitumor effect of Ge-132 is due to increased IFN-γ secretion by activated NK cells and macrophages. Although these studies have documented antitumor effects in vivo or ex vivo, the underlying molecular mechanism has not been clarified. Therefore, we focused on macrophages and investigated the mechanism of activation of macrophages by Ge-132.

This study aimed to investigate the effect and mechanism of the differentiation of M1 macrophages following the intake of Ge-132 (THGP) in vitro or in vivo with a new approach of long-term Ge-132 ingestion. Moreover, we investigated the phenomenon that RAW 264.7 mouse macrophage-like cells cultured with THGP for a long period exhibited increased cytotoxicity toward B16 4A5 melanoma cells in vitro.

## 2. Results

### 2.1. Alteration of RAW 264.7 Cell Properties following Long-Term Culture in Medium Containing THGP

RAW 264.7 cells were cultured for a long period in the presence of 500 μM THGP through repeated passages. As a result, the morphology of the RAW 264.7 cells changed remarkably over time (Figure 1a). We examined the properties of RAW 264.7 cells treated with 500 μM THGP for more than 10 days by evaluating their morphology and proliferation ability. When RAW 264.7 cells were cultured for a short period (less than 10 days) in the presence of 500 μM THGP, no change in morphology was observed. However, after 10 days of culture with THGP, the percentage of cells with a spindle-like morphology was significantly increased by approximately twofold in the THGP-treated group compared to the untreated group (Figure 1b,c). In the subsequent analysis of proliferation, the group treated with 500 μM THGP during the short-term experiment did not show a decrease in cell proliferation compared to the nontreated group (Figure 1d). However, the cells cultured for 10 days in THGP-containing medium showed a decrease in cell growth compared to the control group, and cell growth decreased in a time-dependent manner. Moreover, the growth of RAW 264.7 cells was decreased by treatment with 5000 μM on day 1 (Appendix A). When RAW 264.7 cells were maintained in 50–5000 μM THGP for 10 days, cell growth decreased in a dose-dependent manner (Appendix A). After the cells were treated with 500 μM THGP for a long period, the internal complexity slightly changed, as shown by analysis with flow cytometry (Figure 1e). In addition, from the results of analysis with an auto cell counter, the THGP-treated cells were larger than the control cells (Figure 1f). M1 macrophages exhibit a spindle-like shape, slow cell growth and large size [28,29,30]. The characteristics of RAW 264.7 cells treated with THGP were similar to those of M1 macrophages. Therefore, we suggest that RAW 264.7 cells might differentiate into M1 macrophages following long-term culture in the presence of 500 μM THGP. When RAW 264.7 cells were cultured in the presence of THGP for 10 days or more, they exhibited increased expression of the CD86 molecule and decreased CD206 expression (Figure 1g). The M1/M2 polarization ratio was higher in the THGP-treated cells than in the untreated control cells (Figure 1h). Based on these results, THGP might have induced the macrophages to differentiate toward the M1 phenotype.

### 2.2. THGP Shifted Macrophages toward M1 Polarization

Further analysis of the expression of M1/M2 markers (M1 markers: iNOS, CD80, CD86, TNF-α and IL-1β, and M2 markers: arginase, CD206, CD163, TGF-β and CXCL2) was performed using real-time (RT)–PCR. THGP increased the expression of the M1 markers iNOS by 1.38-fold, CD80 by 2.23-fold, CD86 by 2.23-fold, TNF-α by 1.32-fold and IL-1β by 2.81-fold (Figure 2a) and decreased the expression of the M2 markers arginase by 0.83-fold, CD206 by 0.34-fold, CD163 by 0.69-fold, TGF-β by 0.64-fold and CXCL2 by 0.15-fold (Figure 2b) compared to the levels in the controls. However, when the period of THGP treatment was less than 10 days, the polarization of RAW 264.7 cells did not change considerably compared to that in the nontreated group (Appendix A). These results suggested that macrophages shifted to M1 polarization after long-term THGP treatment (more than 10 days). Subsequently, the expression of the M1 marker CD86 and the M2 marker CD206 was examined using Western blotting. CD86 expression increased approximately 2.5-fold and CD206 expression was significantly decreased, approximately 0.75-fold, after long-term treatment with THGP compared to that in the control cells (Figure 2c–e). Furthermore, RAW 264.7 cells underwent M1 polarization after treatment with lipopolysaccharide (LPS) (Figure 2f,g) and M2 polarization after treatment with the recombinant IL-4 protein (Figure 2h,i). We investigated how THGP affected the differentiation of these cells toward M1 or M2 polarization by performing immunofluorescence staining. Unexpectedly, THGP suppressed the LPS-induced expression of the M1 marker CD86 (Figure 2f). Conversely, THGP did not affect the IL-4-induced expression of the M2 marker CD206 (Figure 2g).

### 2.3. THGP Induces the Differentiation of Macrophages toward M1 Polarization by Activating NF-κB Signaling

First, the intracellular localization of THGP was examined using an isotope microscope to investigate whether THGP was taken up by cells and affected intracellular molecules. The results confirmed that THGP was taken up by RAW 264.7 cells and that elemental germanium (Ge) of THGP was mainly detected in the cytoplasm (Figure 3a). Since THGP was localized in the cells, gene expression was analyzed by performing microarray analysis and compared with that in the control cells and THGP-treated cells. As a result, 217 upregulated genes and 311 downregulated genes showed a greater than 4-fold difference in the THGP-treated cells compared to the control cells (Figure 3b). These differentially expressed genes were analyzed using the enrichment tool Metascape to determine which pathways were affected by THGP. Many of the differentially expressed genes in response to THGP belonged to the group of genes regulated by the transcription factor NF-κB (Figure 3c and Appendix A). Among these genes, 6 genes were upregulated and 16 genes were downregulated by THGP (Figure 3d). Indeed, M1 macrophages activate NF-κB signals [4]. We investigated the p38-NF-κB pathway to confirm the mechanism underlying the THGP-induced transition to M1 polarization. THGP had no effect on the phosphorylation of p38 (Figure 3e,f). However, THGP significantly decreased the level of NF-κB in the cytoplasm (1.0 to 0.35) and increased the amount of NF-kB in the nucleus (0.0 to 1.41) (Figure 3g–i). Furthermore, the differentiation into M1 macrophages induced by THGP was abrogated by the NF-κB inhibitor JSH-23 (Figure 3j,k). Thus, THGP induced the differentiation of RAW 264.7 cells into M1 macrophages by activating NF-κB signaling.

### 2.4. THGP Alters Macrophage Functions Such as Adhesion, Migration and Phagocytosis

Based on the results of the pathway analysis with microarrays, the gene clusters related to cell adhesion, migration and endocytosis were altered by THGP (Figure 4a). First, cell adhesion and migration assays were performed. THGP-treated cells exhibited increased adhesion and decreased migration compared with control cells (Figure 4b–d). Next, a phagocytosis assay was performed. RAW 264.7 cells were maintained in DMEM containing 500 μM THGP for 10 days. After treating RAW 264.7 cells with rabbit IgG FITC-latex beads, the ability of RAW 264.7 cells to phagocytose foreign substances was examined. Notably, 500 μM THGP increased phagocytic activity by 2.5-fold compared to that of the nontreated group (Figure 4e,f).

### 2.5. THGP Increases the Cytotoxicity of RAW 264.7 Cells toward B16 Mouse Melanoma Cells

M1 macrophages have been reported to exhibit cytotoxicity and phagocytose cancer cells. Therefore, we investigated the cytotoxicity and phagocytic activity of RAW 264.7 cells cultured for a long period in the presence of THGP toward cancer cells. First, RAW 264.7 cells and B16 4A5 cells were cocultured in a 96-well plate for 48 h, and cytotoxicity against B16 4A5 cells was evaluated using the MTS assay. No difference in cell growth was observed between RAW 264.7 cells cultured with or without THGP. The absorbance value of cocultures of B16 4A5 cells and RAW 264.7 cells was similar to the addition of their values alone. In contrast, the value of cocultures of B16 4A5 cells and THGP-treated RAW 264.7 cells decreased. The total number of cells cocultured with B16 4A5 cells was lower among the THGP-treated cells than the Ctrl cells (Figure 5a). However, it was not clear whether the difference in cell number between the Ctrl and THGP groups in Figure 5a was due to the decrease in RAW 264.7 cells or B16 4A5 cells. Therefore, we used the luciferase-expressing B16-F10/CMV-LUC#2 mouse melanoma cell line to assess cytotoxicity against B16 melanoma cells by luciferase assays. Both B16 4A5 cells and B16-F10 cells are subspecies of B16 mouse melanoma cells [31,32]. THGP-treated RAW 264.7 cells also were more cytotoxic against B16-F10 cells than untreated control cells (Figure 5b). As shown by a comparison of Figure 5a,b, cytotoxicity was increased approximately 1.3-fold by THGP in both results. Therefore, THGP-treated macrophages were found to exhibit high cytotoxicity to melanoma cells. Next, the ratios of RAW 264.7 cells to B16 4A5 cells were analyzed with flow cytometry when RAW 264.7 cells and B16 4A5 cells were cocultured for 2 h (Figure 5c). RAW 264.7 cells were stained with Green CMFDA dye, and B16 4A5 cells were stained with RED CMFPTX dye. The percentage of B16 4A5 cells decreased by coculture with THGP-treated RAW 264.7 cells compared to control cells (Figure 5d). In addition, PI and Hoechst staining were performed to examine apoptosis in B16 4A5 cells cocultured with RAW 264.7 cells. RAW 264.7 cells were stained green and cocultured with B16 4A5 cells for 2 h. THGP-treated RAW 264.7 cells increased the apoptosis of B16 4A5 cells compared to untreated cells (Figure 2e,f). Therefore, RAW 264.7 cells treated with THGP for 10 days or more were highly cytotoxic to B16 4A5 cells. These data indicate that RAW 264.7 cells treated with THGP for a long period may attack cancer cells more aggressively. Finally, we examined whether the aforementioned results were due to a direct effect of THGP on enhancing the cytotoxicity and phagocytic activity of macrophages toward cancer cells or an indirect effect of cytotoxic cytokines secreted from macrophages (Figure 5g). Thus, the growth rate of B16 4A5 cells was evaluated by performing an MTS assay after incubation with a mixture of the RAW 264.7 cell culture supernatant and normal medium at a 1:1 ratio as conditioned medium. Although THGP was not toxic to B16 4A5 cells, the culture supernatant of RAW 264.7 cells treated with or without THGP inhibited the growth of B16 4A5 cells. However, a significant inhibitory effect on the growth of B16 4A5 cells was not observed for culture supernatants of RAW 264.7 cells (C-CM) and THGP-treated RAW 264.7 cells (T-CM). Therefore, we inferred that THGP might not increase the secretion of cytotoxic cytokines. Based on these results, macrophages may increase direct phagocytosis of cancer cells following treatment with THGP for long periods of time, but cytotoxic humoral factors are unlikely to be involved.

### 2.6. THGP Increases the SIRP-α–CD47 Axis Related to Aggregation and Phagocytosis

Time-lapse movies were recorded using a fluorescence microscope to verify whether the macrophages phagocytosed cancer cells (Movies S1–S4). Phagocytosis of B16 4A5 cells by RAW 264.7 cells treated with or without THGP was observed. However, THGP-treated cells attacked aggressive B16 4A5 cells more intensely than untreated control cells. In addition, THGP-treated RAW 264.7 cells showed a greater increase in the number of aggregated macrophages around cancer cells than untreated control cells (Figure 6a). Furthermore, RAW 264.7 cells and B16 4A5 cells were stained green and blue, respectively, and cocultured for 2 h to investigate the phagocytic activity of RAW 264.7 cells toward B16 4A5 cells. Cells costained with green and blue were recorded as RAW 264.7 cells that had phagocytosed or attacked B16 4A5 cells (Figure 6b). Notably, there was a dramatic increase in costained cells to 63% in the THGP-treated RAW 264.7 cells, compared to 1.4% in the untreated control cells (Figure 6c). Cancer cells express CD47 on the cell surface to avoid phagocytosis by macrophages. Therefore, a neutralizing antibody against CD47 or an inhibitory antibody against SIRP-α (a CD47 recognition receptor expressed on macrophages) has been reported to increase the phagocytic activity of macrophages against cancer cells [7,33]. THGP-treated cells exhibited decreased expression of SIRP-α l compared with control cells, according to the PCR results and Western blot analysis (Figure 6d–f). In addition, the expression of CD47 in B16 4A5 cells was increased by the conditioned medium of RAW264.7 cells (C-CM) but not the supernatant of THGP-treated cells (T-CM) (Figure 6g). In the Western blots, unlike the PCR results, the expression of CD47 in B16 4A5 cells was not changed by C-CM; however, CD47 was expressed at significantly lower levels in cells treated with T-CM than in cells treated with C-CM (Figure 6h,i). Therefore, THGP may increase the recognition of cancer cells and the phagocytic activity of macrophages against cancer cells by suppressing the expression of SIRP-α in macrophages and CD47 in cancer cells.

### 2.7. Ge-132 Induced M1 Macrophage Polarization in Mouse-Derived Primary Intraperitoneal Macrophages In Vivo

Finally, we confirmed whether macrophages differentiated into the M1 or M2 type in the presence of Ge-132 (THGP is a hydrolysate of Ge-132) in vivo. After feeding the mice a diet containing 0.05% Ge-132 for 10, 20 or 30 days and 0.01% Ge-132 for 30 days, experiments were performed using mouse intraperitoneal macrophages. After feeding the mice a diet containing 0.05% Ge-132 for 30 days, the morphology of intraperitoneal macrophages was changed to spindle-like cells (Figure 7a). In addition, the expression levels of the M1 markers CD86 and CD80 were increased by approximately 3-fold and 2-fold, respectively (Figure 7b,c). According to the immunofluorescence staining, the ratio of CD86^+^ M1 macrophages was also increased, and that of CD206^+^ M2 macrophages was decreased (Figure 7d–f). Moreover, phagocytosis of intraperitoneal macrophages derived from mice fed a diet containing 0.05% Ge-132 increased by approximately 1.5-fold compared to that of intraperitoneal macrophages derived from mice fed a control diet (Figure 7g). These results indicate that THGP induces the differentiation of macrophages into M1-type cells not only in vitro but also in vivo.

## 3. Discussion

Immature M0 macrophages differentiate into M1 macrophages or M2 macrophages. M1 macrophages play a role in resistance to bacterial infection. M1 macrophages also function as tumor suppressors in cancer cells through phagocytosis and cytokine secretion. In the present study, we discovered that RAW 264.7 cells differentiated into M1 macrophages upon long-term treatment with THGP. THGP-treated macrophages showed high phagocytic activity against B16 4A5 melanoma cells.

First, the characteristics of RAW 264.7 cells changed significantly following long-term treatment with THGP, including changes in their shape, size and growth rate (Figure 1). In particular, 500 μM THGP had no effect on growth when added for short periods (24–72 h) (Appendix A). However, when the growth was assayed in a THGP-free medium after long-term culture in a THGP-containing medium, significant growth suppression was observed in a culture period-dependent manner. Therefore, we believe that the reason for the decreased proliferation is not cytotoxicity by THGP but the differentiation into M1 macrophages. Indeed, another paper reported that proliferation is reduced when RAW 264.7 cells are differentiated into M1 macrophages by LPS [28]. These characteristics were very similar to those of M1 macrophages reported previously [29]. Indeed, THGP-treated RAW 264.7 cells exhibited increased expression of M1 markers, such as iNOS and CD86, and decreased expression of M2 markers, such as CD206 and CD163 (Figure 2). IFN-γ and LPS are often used to differentiate macrophages into M1 macrophages. In recent years, drugs such as 2-DG and crotamine have also been employed as M1 differentiation inducers in anticancer therapeutic strategies [33,34]. The differentiation of RAW 264.7 cells into M1 macrophages following the addition of M1 macrophage differentiation inducers requires 24 h. In the present study, the differentiation of M0 macrophages into M1 macrophages following THGP treatment required a period of 10 days or more. Even when RAW 264.7 cells were treated with a high concentration of THGP (500 μM), the expression of the M1 marker did not increase until day 10. In other words, when macrophages are induced to differentiate into M1-type macrophages by THGP, the treatment period is more important than the concentration of THGP. As shown in Figure 2f,g, THGP unexpectedly suppressed the induction of M1 macrophages by LPS but not M2 macrophages induced by IL-4. Ge-132, which is a polymer of THGP, has shown anti-inflammatory properties as a treatment for rheumatoid arthritis [35]. Rheumatoid arthritis, in turn, has been reported to be caused by IL-6 and TNF-α secreted from M1 macrophages [36]. In addition, Ge-132 has been shown to suppress the secretion of inflammatory cytokines, such as IL-6 and IL-1β, via p38-NF-κB signaling when primary mouse mammary epithelial cells are treated with LPS [37]. Furthermore, we recently discovered that THGP suppresses inflammasome activation by LPS and ATP stimulation via complexation between THGP and ATP [38]. Therefore, THGP might act as an immunomodulator, not only an M1 differentiation inducer or anti-inflammatory agent.

A total of 217 genes showed greater than 4-fold increases in expression during differentiation, while 311 genes showed greater than 4-fold decreases in expression in the THGP group compared to the control group (Appendix A). Furthermore, 74 genes were upregulated 10-fold, and 103 genes were downregulated 10-fold. The upregulated genes included Hmox1, Ccl3 and CD80. In contrast, Mmp2, Tgfbi and Mmp9 were among the downregulated genes. The microarray data showed that Hmox-1 was upregulated in macrophages treated with THGP. Hmox-1 is expressed at high levels in the livers of rats fed a diet containing Ge-132 compared to the livers of rats fed a control diet for two weeks [39]. Due to the abundance of macrophages known as Kupffer cells in the liver, the expression of Hmox-1 in Kupffer cells might be increased in the livers of rats treated with Ge-132. Conversely, Mmp2, Tgfbi and Mmp9 were downregulated genes. Mmp12 and Mmp2 are involved in the generation of the tumor microenvironment [40]. MMPs promote angiogenesis and cancer metastasis [41]. Tgfbi is a secreted protein induced by transforming growth factor-β (TGF-β), and TGF-β induces epithelial-to-mesenchymal transition in cancer [42]. As a result, THGP suppresses EMT and can prevent cancer metastasis. However, Igf1 and Lyz1, which were reported to be M2 macrophage-related genes, were upregulated, and Ifnar2, which was reported to be related to M1 macrophages, was downregulated [43,44,45]. Thus, macrophages treated with THGP for a long period might not simply differentiate into M1 macrophages. Further research is necessary.

The MAPK, STAT1 and NF-κB pathways are active in M1 macrophages. In contrast, M2 macrophages activate the STAT6 pathway [46]. In this report, THGP directly induced the differentiation of M0 macrophages into M1 macrophages in vitro by activating the NF-κB pathway; however, THGP did not alter the phosphorylation of p38 MAPK, which is upstream of NF-κB (Figure 3). As shown in our previous study, THGP is taken up by normal human dermal fibroblast (NHDF) cells and accumulates in the nucleus [47]. In the present study, THGP was also taken up by RAW 264.7 cells and localized in the cytoplasm (Figure 3a). Moreover, THGP regulates the expression of IL-6, CXCL-2 and NR4A2, which contain binding sites for NF-κB, in NHDFs exposed to oxidative stress [37]. Therefore, THGP might directly or indirectly regulate the nuclear translocation or activity of NF-κB. We must perform further analyses to clarify the mechanism by which THGP induces the differentiation of macrophages toward the M1 phenotype.

In pathway analysis, the expression of many immunity-related genes was altered (Figure 4a). In addition, pathways such as migration, adhesion and endocytosis were also changed. M1 macrophages exhibit higher cell adhesion and lower migration than M2 macrophages through the expression of the myosin protein [48]. Indeed, in our experiments, changes in adhesion and migration were also consistent with the characteristics of M1 macrophages (Figure 4b–d). In addition, based on the microarray results, the expression of Myosin 1D and Myosin F was 26.8- and 18.8-fold higher, respectively, in THGP-treated cells than in Ctrl cells (Appendix A). Therefore, the increase in myosin gene expression may have enhanced the adhesion of THGP-treated cells.

Figure 6 shows that THGP-treated RAW 264.7 cells enhanced phagocytosis via the SIRP-α and CD47 axes. When the cancer cells were treated with macrophage culture supernatant, CD47 expression in cancer cells was increased. A humoral factor (IL-6) from TAMs induces the expression of CD47 in HCC cells [49]. THGP is proposed to have reduced the polarization of M2 macrophages (TAMs) and reduced the secretion of IL-6 by inducing differentiation toward M1 macrophages. Indeed, we clarified the lower expression of IL-6 in THGP-treated macrophages compared to control macrophages using real-time PCR. Therefore, we considered that the expression of CD47 was lower in T-CM than in C-CM due to the difference in the expression of IL-6.

THGP exerts antitumor effects on various cancer types in vivo. The antitumor effect of Ge-132 is believed to induce IFN-γ expression by stimulating NK cells or T cells [50]. In addition, the antitumor effect of macrophages was increased in mice orally administered Ge-132 [27]. However, in these reports, the activation of macrophages by Ge-132 was indicated to be due to the induction of IFN-γ expression resulting from the activation of NK cells. In the present study, the THGP-induced differentiation of macrophages into the M1 phenotype directly activated macrophages rather than via NK cell activation.

Macrophages induced to differentiate into M1 macrophages by THGP exhibited greater phagocytic activity toward FITC-latex beads (used as a foreign substance) than controls (Figure 4e,f). Furthermore, intraperitoneal macrophages were induced to differentiate into M1 macrophages by Ge-132 in vivo. We also showed that these macrophages presented higher phagocytic activity than the control cells (Figure 7g,h). However, when mice ingested a 0.05% Ge-132 diet for 30 days, intraperitoneal macrophages differentiated into the M1 phenotype. When they were fed a 0.05% Ge-132 diet for 10 or 20 days, no change was observed. On the other hand, when a 0.01% Ge-132 diet was provided to mice for 30 days, the number of M1 macrophages increased significantly (Appendix A–c). In other words, these data suggested that Ge-132 must be ingested for a long period to induce the M1 differentiation of macrophages in vivo, similar to the in vitro results.

We also showed that macrophages that were induced to differentiate into M1 macrophages by THGP exhibited increased phagocytic activity against cancers (Figure 5a,b and Appendix A). Furthermore, when culture supernatants from either untreated RAW 264.7 cells or differentiated RAW 264.7 cells treated with THGP were added to B16 4A5 cells, they exerted an antitumor effect on B16 4A5 cells, but the effect was not different from that of the supernatant of RAW 264.7 cells treated with THGP (Figure 5g). In addition, THGP did not directly inhibit B16 4A5 cell growth (Figure 5g). Based on these results, THGP does not affect the production of cytotoxic cytokines by macrophages but increases the phagocytic ability of macrophages by inducing polarization toward the M1 phenotype. In vivo studies have reported that the main factor mediating the antitumor activity of Ge-132 is not NK cells but macrophages or T cells [51]. When Ge-132 treatment was applied together with a T cell- or macrophage-specific blocker, Ge-132 did not exert an antitumor effect. However, the serum from mice treated with Ge-132 showed a tumor-suppressing effect [52]. This effect has been reported to be abolished by treating serum with an anti-IFN-γ antibody. In addition, intraperitoneal macrophages derived from mice orally administered Ge-132 showed high antitumor activity [53]. These reports indicate that Ge-132 activates T cells and induces IFN-γ production, thereby enhancing the cytotoxicity of macrophages against tumors. Ge-132 might certainly exert an antitumor effect as an inducer of IFN-γ production, but the present study also showed for the first time that THGP (hydrolysate of Ge-132) plays a direct role by inducing differentiation into M1 macrophages.

The increase in macrophage phagocytosis is due to the suppression of SIRP-α (a “do not eat me”) expression on the surfaces of macrophages (Figure 6). In fact, the movie shows that macrophages that were induced to differentiate by THGP accumulated around melanoma cells and phagocytosed the cytosol of cancer cells when RAW 264.7 cells and B16 4A5 cells were cocultured (Appendix A). These results revealed a novel mechanism underlying the antitumor effect of THGP.

Propagermanium is another type of THGP polymer crystal, in addition to Ge-132, and both crystals occur in aqueous THGP solution [54]. Propagermanium has been used as a drug to treat hepatitis B and has been reported to inhibit CCR2 [55]. CCR2 is expressed in monocytes, such as macrophages, and monocytes migrate in response to CCL2 secreted from cancer cells [56]. TAMs have an M2 macrophage-like phenotype and generate the tumor microenvironment. Propagermanium inhibits macrophage migration by inhibiting binding between CCR2 and CCL2, thereby suppressing cancer metastasis [57]. Conversely, THGP was shown to induce the differentiation of macrophages into M1 macrophages in the present study, suggesting that THGP might reduce the polarization of M2 macrophages. Therefore, in addition to inhibiting macrophage migration, THGP might suppress cancer growth by reducing the polarization of TAMs. Moreover, propagermanium was recently reported to promote the maturation of NK cells [58]. However, the mechanism of NK cell maturation has not been elucidated. The maturation of NK cells is related to IL-12, IL-18 and IFN-β, which are secreted by M1 macrophages [59]. Therefore, propagermanium might induce NK cell maturation by promoting the differentiation of macrophages into M1 macrophages.

In animal experiments, administration of 0.05% Ge-132 (40–70 mg/kg) for 30 days did not lead to signs of toxicity in mice, such as weight loss. In previous studies, the toxicity of Ge-132 has been investigated in various animal species (mouse, rat and dog), and its safety (extremely low toxicity) has been documented [60,61,62]. The LD50 was also determined in experiments using mice or rats, and its concentration was confirmed to be greater than 11 g/kg following oral administration [63]. Furthermore, Ge-132 has been shown to be nontoxic even when administered at 75 mg/kg in humans in a phase I study, and no toxicity was observed at 1500 mg/day in a case report [35,64]. In addition, Asaigermanium, another name for Ge-132 when used as an ingredient in supplemental health foods in Japan, is produced from the metallic ingot of germanium, while the similar Ge-132 is produced from toxic GeO_2_. Therefore, Asaigermanium is not contaminated with GeO_2_ from the base material. Only Asaigermanium passed the safety assessment of the Japanese Generally Recognized as Safe (GRASS) guidelines for use as a food ingredient in 2019. Therefore, Ge-132 (Asaigermanium) can be safely used as a medicine or health food with protective effects on tumor development.

In summary, macrophages were shown to differentiate into the M1 phenotype following long-term culture in the presence of THGP. Macrophages treated with THGP for a long period showed higher phagocytic activity and cytotoxicity against cancer cells than control macrophages. The mechanism by which THGP promoted cancer cell phagocytosis might involve the suppression of SIRP-α and CD47 (two “do not eat me” signals) expression. However, why THGP induces decreased SIRP-α expression is still unclear, but these results suggest that THGP might be useful as a novel anticancer agent. Alternatively, Ge-132, commonly known as “Asaigermanium”, is used as a food health supplement in Japan and some Asian countries. Therefore, daily ingestion of Ge-132, which has been confirmed to be highly safe, as a health food is expected to activate macrophages, prevent carcinogenesis and protect against diseases and infection. Such usage of Ge-132 may be also a new approach for cancer immunotherapy. Thus, we should investigate the antitumor effect of macrophages obtained after M1 differentiation induced by THGP in vivo.

## 4. Materials and Methods

### 4.1. Cell Culture

RAW 264.7 cells and B16 4A5 cells were provided by Riken Cell Bank (Riken BRC, Ibaraki, Japan). B16-F10/CMV-LUC#2 cells were provided by the JCRB Cell Bank (JCRB Cell Bank, Osaka, Japan). The cells were cultured with Dulbecco’s modified Eagle’s medium (DMEM) (Nissui Pharmaceutical Co., Ltd., Tokyo, Japan) supplemented with 10% fetal bovine serum (FBS) (Biosera Europe, Nuaille, France) at 37 °C in the presence of 5% CO_2_. When the cells reached subconfluence, they were removed with a scraper or by treatment with 0.25% trypsin/1 mM EDTA (Nacalai Tesque, Inc., Kyoto, Japan) and seeded at a ratio of 1:4–8 every 2–3 days. In Figure 1b,c, Figure 2a,b and Figure 4e,f, RAW 264.7 cells were cultured with 500 μM THGP for 10 days with repeated passages. In the other experiments, RAW 264.7 cells were cultured with 500 μM THGP for more than 10 days.

### 4.2. Cell Growth Assay (MTS Assay)

Cells were seeded in a 96-well plate at a density of 5 × 10^3^ cells/well and cultured with THGP-free medium. After 24 h or 48 h, cell numbers were analyzed using a CellTiter 96^®^ AQueous One Solution Cell Proliferation Assay kit (Promega Corp., Madison, WI, USA) according to the manufacturer’s protocol.

### 4.3. Analysis of Cell Size and Internal Complexity

Cells were collected using a cell scraper and adjusted to a density of 1 × 10^6^ cells/200 μL. Then, the cells were fixed with 1% paraformaldehyde and 1% goat serum albumin (Abcam Ltd., Cambridge, UK) in PBS. Cell size and internal complexity were analyzed using an Accuri C6 flow cytometer (BD Life Sciences, Franklin Lakes, NJ, USA) and BD Accuri C6 software, version 1.0.264.21 (BD Biosciences). Cell size was also analyzed using a Countess^®^ II FL auto cell counter (Invitrogen, Life Technologies Corp., Carlsbad, CA, USA).

### 4.4. Immunofluorescence Staining

Cells were seeded on a cover glass (Matsunami Glass Ind., Ltd., Osaka, Japan) in a 6-well plate at a density of 1 × 10^6^ cells/well. After 24 h, the cells were immobilized with 4% paraformaldehyde in PBS (FUJIFILM Wako Pure Chemical Corporation, Osaka, Japan) and subjected to permeabilization with 0.2% Triton X-100 (Nacalai Tesque, Inc.) for 10 min at room temperature (RT). The cells were blocked with 1.5% bovine serum albumin (BSA) (FUJIFILM Wako Pure Chemical Corporation) in phosphate-buffered saline containing 0.2% Triton X-100 for 30 min. The primary antibody incubation was performed overnight at 4 °C, and the secondary antibody (Abcam Ltd., Cambridge, UK) was incubated with cells at RT for 1 h. The nuclei were stained with DAPI (Dojindo Laboratories, Kumamoto, Japan) and observed with a BZ-X810 fluorescence microscope (KEYENCE Corporation, Osaka, Japan). Anti-B7-2 (CD86) (sc-28347) (Santa Cruz Biotechnology, Inc., Santa Cruz, CA, USA), anti-mannose receptor (CD206) (ab64693) (Abcam Ltd.) and anti-α-tubulin (ab52866) (Abcam Ltd.) antibodies were used as primary antibodies. The CD86 antibody was diluted 1:50, and the CD206 antibody was diluted 1:500 in blocking buffer.

### 4.5. Quantitative Polymerase Chain Reaction (PCR)

RAW 264.7 cells or B16 4A5 cells were seeded in a six-well plate at a density of 5 × 10^5^ cells/well. After 24 h, total RNA was extracted using Isogen reagent (Nippon Gene, Inc., Tokyo, Japan) according to the manufacturer’s recommended protocol. The extracted RNA was reverse transcribed using SuperScript III (Invitrogen) with 1 μg of template. When analyzing mouse peritoneal macrophages, the RNAqueous™-Micro Total RNA Isolation Kit (Invitrogen) and SuperScript IV VILO Master Mix (Invitrogen) were used. Real-time PCR consisted of 2 steps (40 cycles of 95 °C for 5 s and 60 °C for 30 s) and was performed using TB Green Premix Ex Taq II (Tli RNaseH Plus) (Takara Bio, Inc., Otsu, Japan) and a LightCycler 96 (Roche Diagnostics GmbH, Mannheim, Germany). The primers used for PCR are shown in Appendix A, and RPS18 was used as an internal control.

### 4.6. Western Blotting

Cells were seeded at a density of 5 × 10^6^ cells/6-cm dish. After 24 h, the cells were dissolved in low-salt buffer. The supernatant was recovered as the cytoplasmic fraction after centrifugation at 1500× *g*. The precipitate was dissolved in high-salt buffer, and the supernatant was collected as the nuclear fraction after centrifugation at 15,000× *g*. Both fractions were quantified using the Bradford method and adjusted to a concentration of 0.5 μg/μL. SDS–PAGE was performed using 5 μg of protein, after which the proteins were transferred to PVDF membranes. When detecting phosphorylated proteins, 5% BSA in TBS-T (Tris-buffered saline with 0.01% Tween-20) was used as a blocking buffer. When detecting other proteins, 5% skim milk in TBS-T was used as blocking buffer. First, antibodies were dissolved in blocking buffer and incubated with the membrane overnight at 4 °C. Secondary antibodies were dissolved in blocking buffer and incubated with the membrane for 1 h at room temperature. Anti-B7-2 (1:200), anti-mannose receptor (1:500), p-38 (Santa Cruz) (1:200), p-p38 (Cell Signaling Technology Inc., Danvers, MA) (1:1000), NF-κB (Santa Cruz) (1:500), NF-κB (Cell Signaling) (1:1000), SIRP-α (Abcam) (1:1000), CD47 (Abcam) (1:1000), β-actin (Abcam) (1:3000) and lamin B1 (Abcam) (1:1000) antibodies were used as primary antibodies.

### 4.7. Induction of M1 or M2 Polarization

Cells were seeded in a six-well plate at a density of 1 × 10^5^ cells/well. After 24 h, the cells were treated with LPS (Sigma-Aldrich Corp., St. Louis, MO, USA) at 100 ng/µL or IL-4 (BioLegend, San Diego, CA, USA) at 10 ng/μL to induce M1 or M2 polarization. After 48 h, the cells were fixed with 4% paraformaldehyde in PBS and stained with a CD86 or CD206 antibody.

### 4.8. Isotope Microscopy

The isotope microscopy procedures were described in a previous study [47]. Briefly, RAW 264.7 cells were seeded on silicon wafers placed in 35-mm dishes at a density of 2.0 × 10^5^ cells/wafer and cultured for 1 day with 10 mM THGP. After fixing the cells on silicon wafers with resin, Ge (elemental germanium), P (elemental phosphorus) and CN (elemental carbon and nitrogen) levels in cells were analyzed with a Hokudai isotope microscope system (CAMECA IMS 1270 and SCAPS ion imager at Hokkaido University).

### 4.9. Microarray Analysis

Total RNA was extracted from RAW 264.7 cells treated with or without Isogen reagent (Nippon Gene, Inc., Tokyo, Japan), and the quality was evaluated by measuring the absorbance at wavelengths of 260/280 nm. The whole transcriptome expression profile was determined using a microarray Affymetrix Clariom S mouse array and GeneChip WT Plus Reagent Kit (Thermo Fisher Scientific Inc., Madison, MA, USA) according to the manufacturer’s instructions. Briefly, 100 ng of total RNA was used to generate cDNAs, and fragmented and labeled cDNAs were hybridized to a Clariom S mouse array for 16 h at 45 °C. The arrays were washed, stained using GeneChip Fluidics Station (Thermo Fisher Scientific) and then scanned using the Affymetrix GeneChip Scanner 3000 7G (Thermo Fisher Scientific). CEL intensity files were generated with Affymetrix GeneChip Command Console software (Thermo Fisher Scientific). The CEL files were analyzed and normalized using the RMA-TMA method with Transcriptome Analysis Console (TAC) software, version 4.0 (Thermo Fisher Scientific). Gene expression profiles were compared between the nontreated control group (n = 4) and the THGP-treated group (n = 5) using TAC software with the thresholds of *p* < 0.01 and fold changes <−4 or >4. The filtered, differentially expressed genes were functionally enriched using the Metascape web tool (https://metascape.org). I have accessed on 29 January 2021 in Metascape. In the pathway analysis, the dataset was analyzed for GO terms.

### 4.10. NF-κB Inhibitor Test

Cells were cultured with 500 μM THGP in the presence or absence of the NF-κB inhibitor JSH-23 (5 μM) for 10 days with repeated passages. After culture for 10 days, proteins were collected, and Western blotting was performed.

### 4.11. Cell Adhesion Assay

Cells were seeded in a collagen-I-coated 96-well plate at a density of 2 × 10^4^ cells/well. After 30 min, nonadherent cells were removed by washing three times with PBS (-). The number of cells was evaluated using the MTS assay. The percentage of adherent cells in each experimental group was determined by counting the total number of cells per well (floating plus adherent cells).

### 4.12. Cell Migration Assay

Cells were seeded in a 24-well plate at a density of 2 × 10^5^ cells/well. After 24 h, the cells were scratched with a 1000-μL tip. After 24 h, the closed area was analyzed using a BZ-X800 analyzer (KEYENCE).

### 4.13. Phagocytosis Assay

Cells were seeded in a six-well plate at a density of 5 × 10^5^ cells/well. After 24 h, their phagocytic activity was evaluated with a phagocytosis assay kit (Cayman Chemical Company, Ann Arbor, MI, USA). FITC beads were diluted 1:200 with DMEM, and the cells were treated with the beads for 1 h. The nuclei were stained with Hoechst 33452 (Dojindo Laboratories) and observed with a fluorescence microscope. The analyzed cells appear in Gate R1 in Figure 1e.

### 4.14. Coculture Model with MTS Assays

RAW 264.7 cells and B16 4A5 cells were seeded in 96-well plates at densities of 5 × 10^3^ cells/well and 2.5 × 10^3^ cells/well, respectively, and monocultured or cocultured for 48 h in THGP-free medium. Cell numbers were assessed using the MTS assay. The results are presented in Figure 5a and show the absorbance of RAW 264.7 cells or B16 4A5 cultured alone or cocultured, as determined using the MTS assay.

### 4.15. Coculture Model with Luciferase Assays

RAW 264.7 cells and B16-F10/CMV-LUC#2 cells were seeded in 96-well black plates at densities of 5 × 10^3^ cells/well and 2.5 × 10^3^ cells/well, respectively, and monocultured or cocultured for 48 h in THGP-free medium. B16-F10 cell numbers were assessed by luciferase assays. After the medium was removed, 100 μL of ONE-Glo™ Luciferase Assay System buffer (Promega) and 100 μL of DMEM were added to the 96-well plates. Luminescence intensity was then measured by ARVO™ X3 (PerkinElmer, Inc., Waltham, MA, USA).

### 4.16. Coculture Model with Fluorescence Staining

RAW 264.7 cells were stained with CellTracker Blue CMAC Dye (Thermo Fisher Scientific Inc.) and seeded in 6-well plates at a density of 1 × 10^6^ cells/cover glass. After 24 h, B16 4A5 cells were stained with CellTracker Green CMFDA Dye (Thermo Fisher Scientific Inc.), and 5 × 10^5^ cells were seeded over RAW 264.7 cells. The cells were incubated for 2 h at 37 °C. After washing the surface with PBS (-) twice, the cells were fixed with 4% paraformaldehyde in PBS and observed with a fluorescence microscope. Images were analyzed using NIS-Elements software (Nikon, Tokyo, Japan). RAW 264.7 cells that phagocytosed B16 4A5 cells were identified as double-positive cells with both green and blue staining. For the flow cytometry analysis, RAW 264.7 cells were stained with CellTracker Green CMFDA dye and seeded at a density of 2 × 10^6^ cells/6-well plate. After 24 h, B16 4A5 cells were stained with CellTracker RED CMFPTX Dye (Thermo Fisher Scientific Inc.), and 1 × 10^6^ cells were seeded on RAW 264.7 cells. Cells were incubated for 2 h at 37 °C, collected with a cell scraper and fixed with 1% paraformaldehyde and 1% goat serum albumin (Abcam) in PBS. The percentages of B16 4A5 cells and RAW 264.7 cells were analyzed using flow cytometry.

### 4.17. Coculture Model with PI and Hoechst Staining

RAW 264.7 cells were stained with CellTracker Green CMFDA dye and seeded in 6-well plates at a density of 1 × 10^6^ cells/well. After 24 h, B16 4A5 cells were seeded over RAW 264.7 cells at a density of 5 × 10^5^ cells/well. Cells were incubated for 2 h at 37 °C, and then all cells, including attached cells and floating cells, were collected with a scraper and centrifuged at 3000 rpm for 5 min. The collected cells were stained with propidium iodide (DOJINDO LABORATORIES) and Hoechst 33258 for 40 min at 37 °C and observed with a fluorescence microscope. Apoptotic B16 4A5 cells were defined as blue^+^ green^−^ red^+^ cells.

### 4.18. Collection of Mouse Peritoneal Macrophages

Male ICR mice (Charles River Laboratories Japan, Inc., Yokohama, Japan) were used. The mice were fed the control diet or a 0.05% Ge-132 diet for 10, 20 or 30 days or a 0.01% Ge-132 diet for 30 days. Then, intraperitoneal macrophages were collected using PBS. After hemolysis, 1/8 of the total cells were seeded in an 8-well glass plate (Matsunami Glass Ind., Ltd.) for the phagocytosis assay and immunofluorescence staining, and 1/2 of the total cells were seeded in a 48-well plate for RT–PCR. After 24 h, the cells were washed with PBS, and the adherent cells were used as macrophages. The animal experiments were all conducted at Asai Germanium Research Institute Co., Ltd., Hakodate, Japan, according to the guidelines provided by the Ethical Committee of Experimental Care, which were based on public guidelines set by the Japanese Ministry of Education, Culture, Sports, Science and Technology. The studies were individually approved and performed according to the ethical guidelines of the judging committee at Asai Germanium Research Institute Co., Ltd.

### 4.19. Conditioned Medium

RAW 264.7 cells were seeded at a density of 1 × 10^6^ cells/10-cm dish and cultured for 3 days, after which the supernatant was collected. The supernatant and the normal medium were mixed at a 1:1 ratio. B16 4A5 cells were cultured with conditioned medium for 3 days, and RNA was collected with Isogen reagent or analyzed with the MTS assay.

### 4.20. Video Recording of Cocultured RAW 26 4.7 Cells and B16 4A5 Cells

RAW 264.7 cells were seeded in a 6-well plate at a density of 1.5 × 10^5^ cells per well. After 24 h, B16 4A5 cells were stained with CellTracker RED CMFPTX Dye and seeded at a density of 5 × 10^4^ cells/well. Time-lapse video recordings were obtained at 10-min intervals for 72 h with a BZ-X810 microscope (KEYENCE Corporation, Osaka, Japan). Fluorescence was visualized at the beginning and end of the video.

### 4.21. Statistical Analysis

Statistical analyses were performed with Statcel—the Useful Addin Forms in Excel, 4th edition (OMS Publishing Co., Ltd., Tokyo, Japan). The data were analyzed using Student’s two-tailed t test for two groups. Data were compared among more than two groups using one-way ANOVA and Dunnett’s test. *p* < 0.05 was considered to indicate a statistically significant difference.

## Figures and Tables

**Figure 1 ijms-24-01885-f001:**
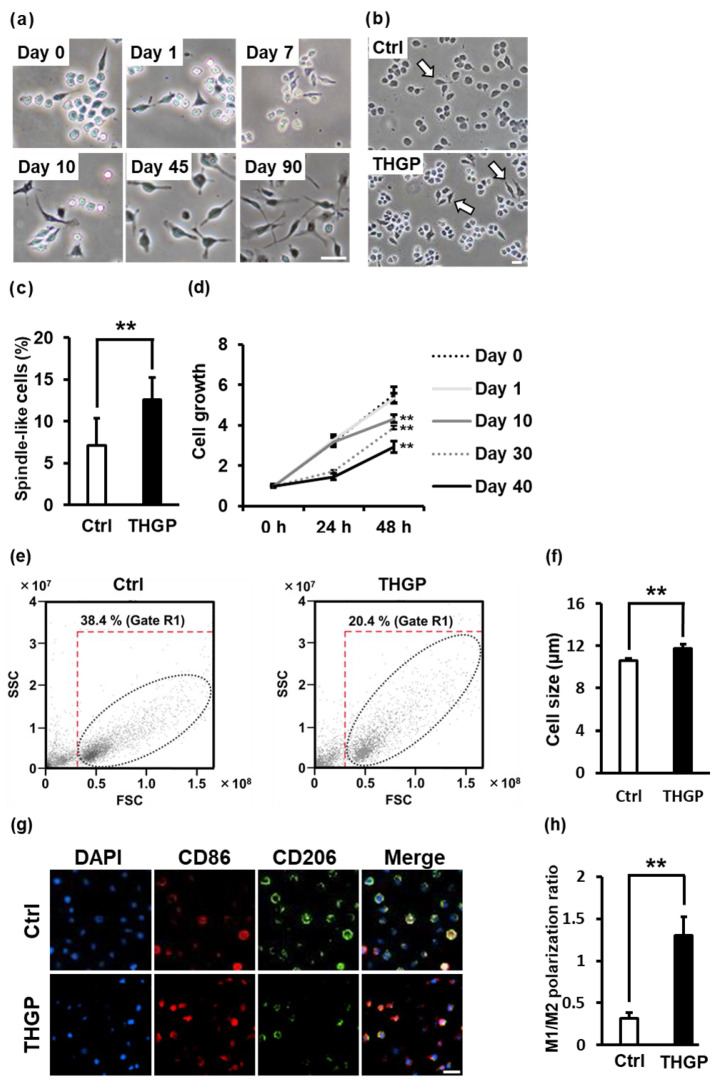
Morphological changes in macrophages after long-term culture with THGP. (**a**) Micrographs show RAW 264.7 cells cultured for 0, 1, 7, 10, 45 and 90 days in medium supplemented with or without 500 μM THGP through repeated passages. Scale bars: 20 μm. (**b**) Micrographs show RAW 264.7 cells cultured for 10 days in medium supplemented with or without 500 μM THGP. Arrows indicate spindle-like cells. Scale bars: 20 μm. (**c**) The percentage of spindle-like cells was calculated. (**d**) RAW 264.7 cells were maintained in medium containing 500 μM THGP for 0, 10, 20 or 40 days with repeated passages. Their cell proliferation rate was assessed using an MTS assay. (**e**) The graphs show side scatter (SSC-A) and forward scatter (FSC-A) of RAW264.7 cells analyzed using flow cytometry after treatment with or without THGP. (**f**) Their cell sizes were analyzed with a cell counter. (**g**) Photographs show the immunofluorescence staining for CD86 (green) or CD206 (red) and the cell nucleus stained with 4′,6-diamidino-2-phenylindole (DAPI; blue) in RAW 264.7 cells cultured in the absence or presence of 500 μM THGP for 10 days or more. Scale bars: 20 μm. (**h**) The ratio of CD86-positive/CD206-positive cells is shown. The results are presented as the means ± S.D. (n = 6). ** *p* < 0.01 compared with the controls.

**Figure 2 ijms-24-01885-f002:**
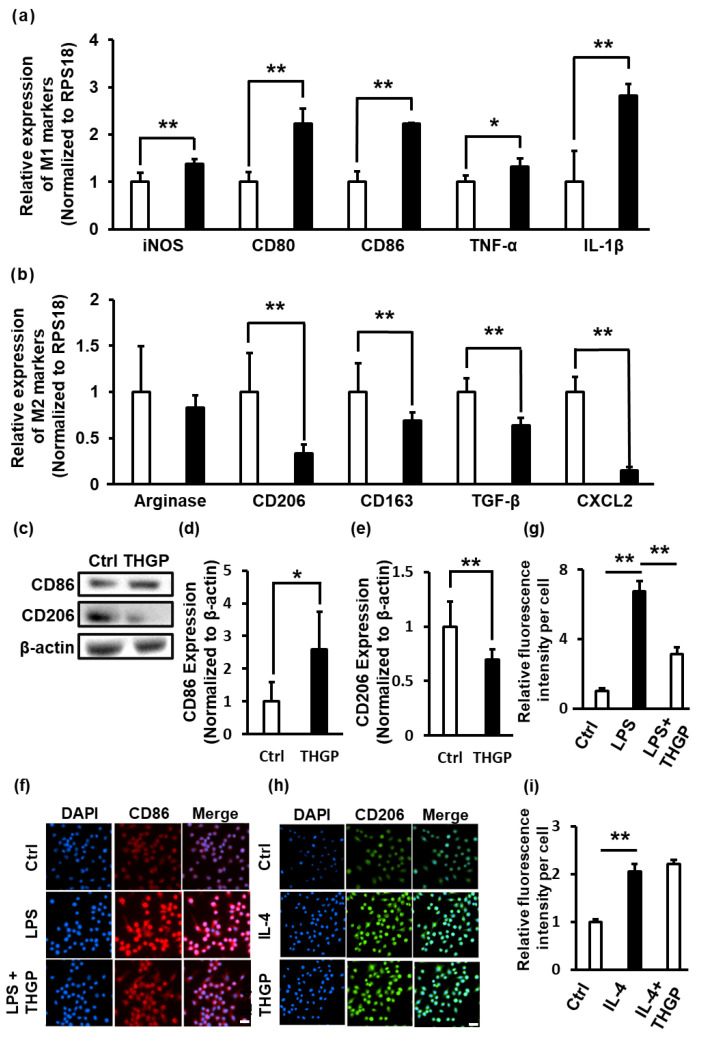
Effect of THGP on the M1/M2 polarization of RAW 264.7 cells during long-term culture. (**a**) The expression of M1 markers was analyzed using RT–PCR. RAW 264.7 cells were cultured for more than 10 days in the absence (Ctrl, white bar) or presence (THGP, black bar) of 500 μM THGP. (**b**) The expression of M2 markers was analyzed using RT–PCR. RPS18 was used as an internal control. (**c**–**e**) The expression levels of CD86 and CD206 were determined using Western blotting and quantified. β-actin was used as an internal control. (**f**) CD86 expression was observed under a fluorescence microscope after treatment with LPS to induce differentiation into M1 macrophages. Ctrl: untreated control cells, LPS: cells treated with 100 ng/mL LPS, LPS + THGP: cells treated with 100 ng/mL LPS and 500 μM THGP. The nucleus was stained with DAPI (blue), and CD86 is shown in red. Scale bars: 20 μm. (**g**) The fluorescence intensity of CD86 per cell was quantified in (**f**). (**h**) CD206 expression was observed under a fluorescence microscope after treatment with IL-4 to induce differentiation into M2 macrophages. Ctrl: untreated control cells, IL-4: cells treated with 10 ng/mL IL-4, IL-4 + THGP: cells treated with 500 μM IL-410 ng/mL and THGP. The nucleus was stained with DAPI (blue), and CD206 is shown in green. Scale bars: 20 μm. (**i**) The fluorescence intensity of CD206 per cell was quantified in (**i**). The results are presented as the means ± S.D. (n = 6). * *p* < 0.05 and ** *p* < 0.01 compared with the control.

**Figure 3 ijms-24-01885-f003:**
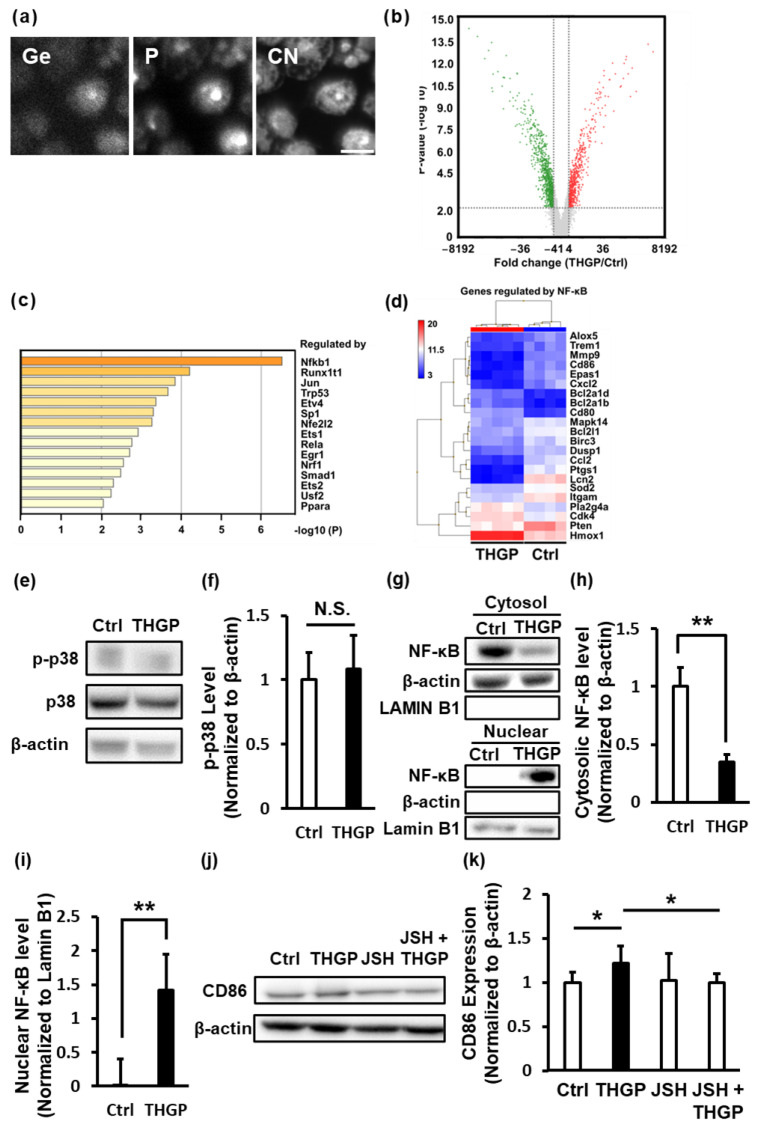
Analysis of the mechanism of M1 macrophage differentiation induced by THGP. (**a**) The panels show RAW 264.7 cells treated with 500 μM THGP analyzed with an isotope microscope. Ge (elemental germanium) represents THGP, P (elemental phosphorus) represents the nucleus and CN (elemental carbon and nitrogen) represents the cytoplasm. Scale bars: 10 μm. (**b**) Volcano plot of the expression levels and *p* values in RAW 264.7 cells treated with THGP compared with untreated RAW 264.7 cells (Ctrl). Red plots represent the genes with a greater than 4-fold increase in expression in THGP-treated cells compared to the controls, and green plots represent the genes with a less than 4-fold decrease in expression in THGP-treated cells compared to the controls. (**c**) The enrichment analysis using Metascape showing the possible transcription factors that regulate the genes with differential expression in THGP-treated cells compared with Ctrl cells. (**d**) The expression heatmaps of the genes in the NF-κB pathway in cells cultured in the presence or absence of THGP. (**e**,**f**) The levels of p-38 and p-p38 were determined using Western blotting and quantified. β-actin was used as an internal control. (**g**–**i**) Levels of NF-κB in the cytosol and nucleus were quantified using Western blotting. β-actin was used as an internal control for the cytosol, and lamin B1 was used as an internal control for the nucleus. The results are presented as the means ± S.D. (n = 6). (**j**,**k**) The expression of CD86 on RAW 264.7 cells was determined using Western blotting after treatment with 500 μM THGP with or without 5 μM JSH-23, an NF-κB inhibitor, for 10 days. * *p* < 0.05 and ** *p* < 0.01 compared with the controls. N.S. = Not significant.

**Figure 4 ijms-24-01885-f004:**
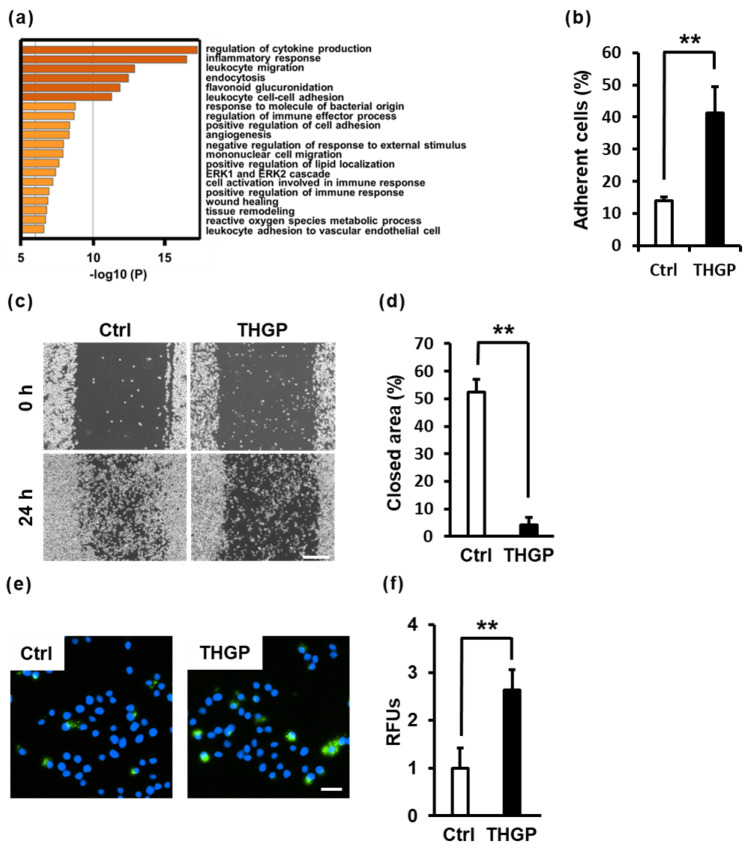
Evaluation of the functions of RAW 264.7 cells treated with THGP. (**a**) The graph shows the results of the pathway analysis of differentially expressed genes in THGP-treated cells compared with the control group using Metascape. (**b**) The graph shows the percentage of adherent cells among the total cells on the collagen-coated plate within 30 min after seeding. (**c**) The photographs show the migration of cells at 0 h and 24 h after scratching the cells. Scale bars: 200 μm. (**d**) The graph shows the rate of migration of cells in (**c**). (**e**) After RAW 264.7 cells were cultured for 10 days with 500 μM THGP, a phagocytosis assay was performed. In the photographs, green and blue signals represent phagocytosed FITC-beads and the cell nucleus (Hoechst 33452), respectively. Scale bars: 20 μm. (**f**) The fluorescence intensity was analyzed in the photographs shown in (**e**). The results are presented as the relative total intensity obtained from the total cells and are reported as the means ± S.D. (n = 6). ** *p* < 0.01 compared with the controls.

**Figure 5 ijms-24-01885-f005:**
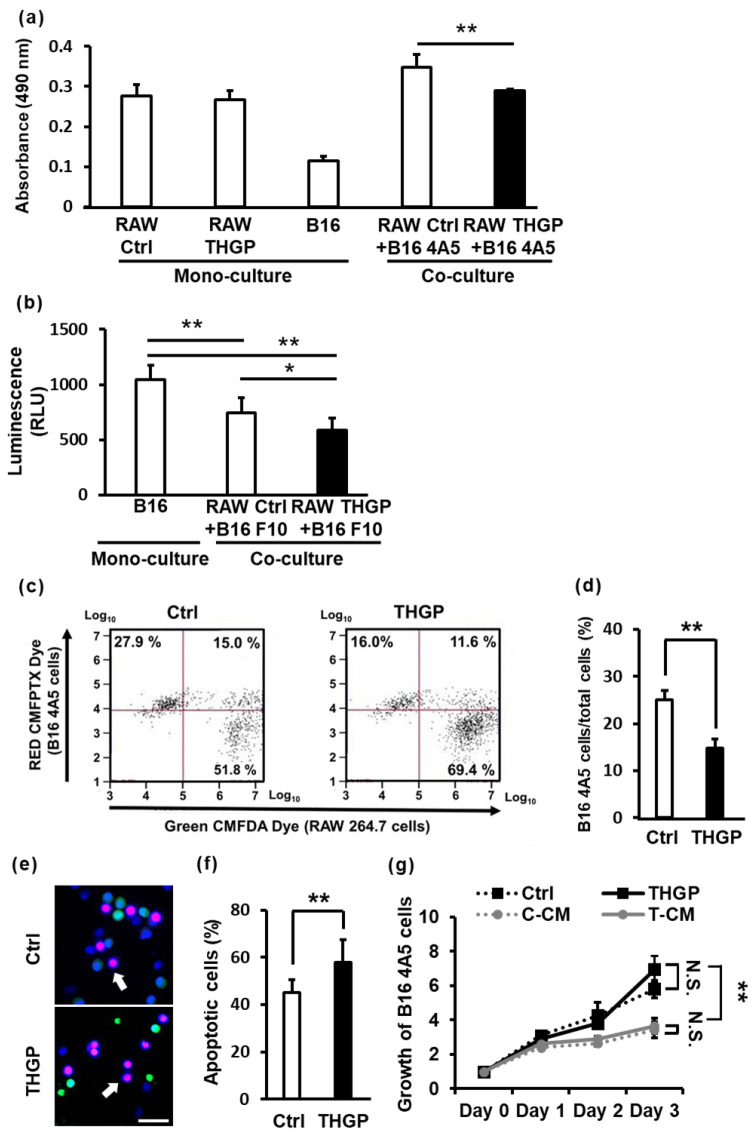
THGP increases the cytotoxicity of RAW 264.7 cells toward B16 4A5 melanoma cells. (**a**) The cytotoxicity of RAW 264.7 cells against B16 4A5 cells was evaluated via MTS assay. (**b**) The cytotoxicity of RAW 264.7 cells against B16-F10 LUC #2 cells was evaluated using a luciferase assay. (**c**) After staining with the dye CMFPTX, B16 4A5 cells (red) and RAW 264.7 cells (green) were cocultured and analyzed using flow cytometry. (**d**) Then, the percentages of B16 4A5 cells were determined. (**e**) The image shows apoptotic cells analyzed with PI and Hoechst staining. Total cells were stained blue with Hoechst 33258. RAW 264.7 cells were stained green with Cell Tracker Green. Apoptotic cells were stained red with propidium iodide. Apoptotic B16 4A5 cells are shown as blue^+^ green^−^ red^+^ cells (shown by white arrows). Live B16 4A5 cells are shown as blue^+^ green^−^ red^−^ cells. Apoptotic RAW 264.7 cells are shown as blue^+^ green^+^ red^+^ cells. Live RAW 264.7 cells are shown as blue^+^ green^+^ red^−^ cells. Scale bars: 20 μm. (**f**) The graph shows the percentage of apoptotic B16 4A5 cells cocultured with control RAW 264.7 cells or THGP-treated RAW 264.7 cells for 10 days or more. (**g**) The figure shows the growth rate of B16 4A5 cells cultured in conditioned medium supplemented with or without 500 μM THGP and in the mixture of the RAW 264.7 culture supernatant supplemented with or without 500 μM THGP and normal medium at a 1:1 ratio determined using the MTT assay. Ctrl: normal medium; THGP: normal medium containing 500 μM THGP; C-CM: untreated RAW 264.7 cell culture supernatant; T-TM: THGP-treated RAW 264.7 cell culture supernatant. n = 6. * *p* < 0.05 and ** *p* < 0.01 compared with the control. N.S.= Not significant.

**Figure 6 ijms-24-01885-f006:**
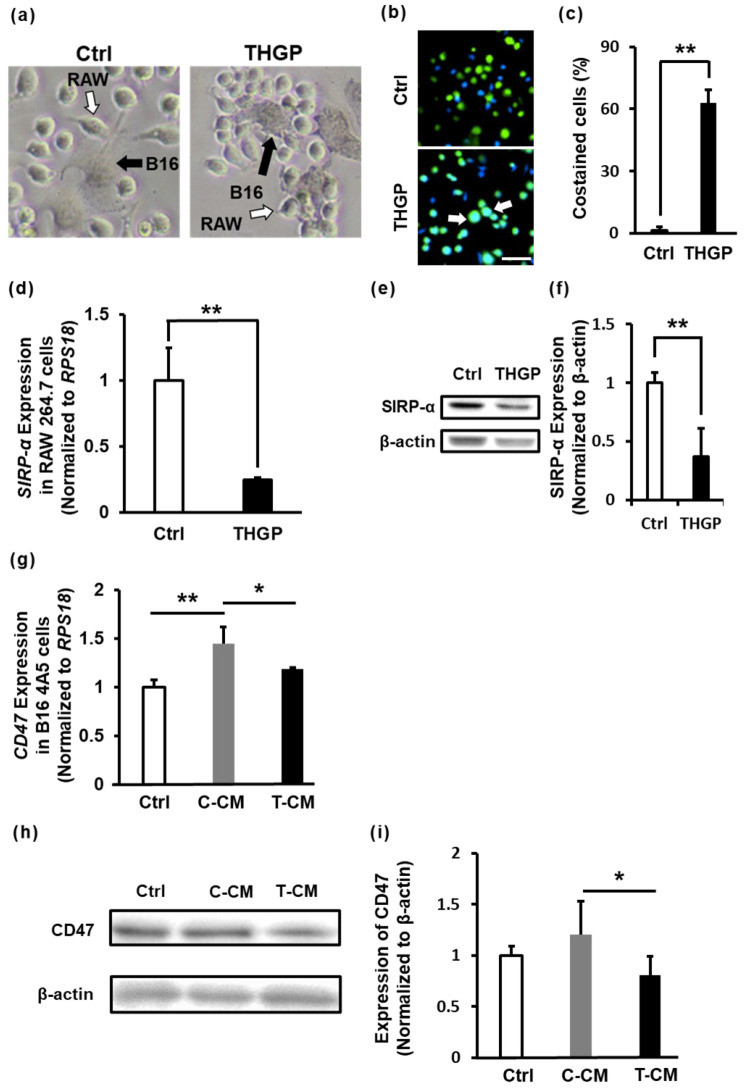
Expression of genes related to phagocytosis in macrophages. (**a**) Bright-field images show the coculture of B16 4A5 cells and RAW 264.7 cells treated with or without THGP for 10 days or more. The images were taken at 20× magnification. (**b**) Fluorescence images obtained after the staining of RAW 264.7 cells (green) and B16 4A5 cells (blue). Ctrl represents B16 4A5 cells cocultured with THGP-untreated cells. THGP represents B16 4A5 cells cocultured with RAW 264.7 cells cultured with 500 μM THGP-containing medium for 10 days or more. Scale bars: 20 μm. (**c**) The percentage of B16 4A5 cells phagocytosed by macrophages was calculated. Phagocytosed cells were defined as those double positive for green and blue staining (indicated by the white arrows in Figure 6b). (**d**) SIRP-α expression in RAW 264.7 cells treated with or without THGP was analyzed using RT–PCR. (**e**,**f**) SIRP-α expression was determined using Western blotting and quantified in RAW 264.7 cells treated with or without THGP. β-actin was used as an internal control. (**g**–**i**) B16 4A5 cells were cultured in monoculture (Ctrl), C-CM or T-CM. Then, CD47 expression was analyzed using RT–PCR (**g**). CD47 expression was determined using Western blotting and quantified. β-actin was used as an internal control (**h**,**i**). * *p* < 0.05 and ** *p* < 0.01 compared with the controls.

**Figure 7 ijms-24-01885-f007:**
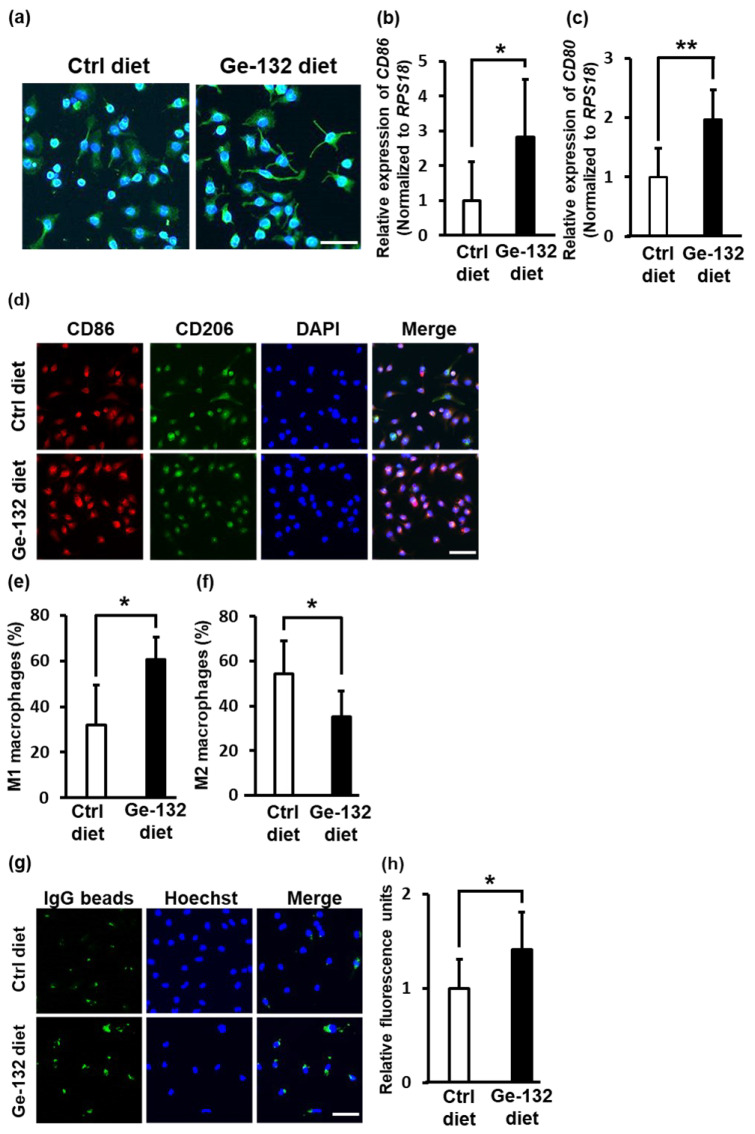
Effect of feeding mice a Ge-132 diet on M1 macrophage polarization in murine intraperitoneal macrophages in vivo. (**a**) Fluorescence photographs reveal the morphology of primary intraperitoneal macrophages in mice fed a control or 0.05% Ge-132 diet for 30 days. Green represents the cytoskeleton stained with α-tubulin, and blue represents the nucleus stained with DAPI. RT–PCR was performed to determine the expression of the M1 markers CD86 (**b**) or CD80 (**c**) in primary intraperitoneal macrophages in mice fed a control diet or 0.05% Ge-132 diet for 30 days. (**d**) Photographs showing immunofluorescence staining for CD86 (green) and CD206 (red) along with DAPI (blue) in primary intraperitoneal macrophages from mice fed a control diet or 0.05% Ge-132 diet for 30 days. (**e**) M1 macrophage cells with high expression of CD86 and low expression of CD206 in (**d**). (**f**) M2 macrophage cells with low expression of CD86 and high expression of CD206 in (**d**). (**g**) A phagocytosis assay was performed. The green signal represents phagocytosed FITC-beads, and the blue signal represents the cell nucleus (Hoechst 33452). (**h**) Analysis of the fluorescence intensity in (**g**). The results show the relative total intensity obtained from the total cells. n = 6. ** *p* < 0.01 and * *p* < 0.05 compared with the controls. Scale bars: 20 μm.

## Data Availability

The data are available upon request from the authors.

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
