# Peer review of "Organogermanium THGP Induces Differentiation into M1 Macrophages and Suppresses the Proliferation of Melanoma Cells via Phagocytosis"

_ijms, 2023, doi:10.3390/ijms24031885_

Round 1

Reviewer 1 Report (Previous Reviewer 2)

I tried to head the authors to the notion that it is Activin A, but not TGFb1, the TGFb-family protein expressed in human melanoma. Although they did not change the cytokine in the main text, at least it is properly cited now.

Author Response

To Reviewer 1

Thank you for your kind comments. I believe the manuscript was improved by the modifications. The responses to the comments are provided below.

I tried to head the authors to the notion that it is Activin A, but not TGFb1, the TGFb-family protein expressed in human melanoma. Although they did not change the cytokine in the main text, at least it is properly cited now.

A)Thank you for sharing important insights about TAMs.

I apologize for changing the contents that you did not address. I changed TGF-B to ACTIVIN A and changed the content to match the citation (Lines 58-60).

Reviewer 2 Report (New Reviewer)

It is a well-designed and written paper and carries considerable merits, but some points need an explanation before the manuscript is ready for publication:

The introduction and discussion should be focused more on the observations and novelty of this study. More concluding remarks must be also added.

Author Response

To Reviewer 2

Thank you for your kind comments. I believe the manuscript was improved by the modifications. The responses to the comments are provided below.

Comment 1

It is a well-designed and written paper and carries considerable merits, but some points need an explanation before the manuscript is ready for publication:

The introduction and discussion should be focused more on the observations and novelty of this study. More concluding remarks must be also added.

A) Thank you for your advice. The introduction and discussion have been revised to describe the novelty of this study.

Reviewer 3 Report (New Reviewer)

In this manuscript, the authors  showed that THGP could promote the macrophages polarization to M1 in vitro and in vivo. In addition, they showed that THGP could decrease NF-κB expression and SIRP-α expression in vitro.

The authors should address following questions to strengthen the manuscript.

1.  In Figure 1d, the authors showed that the cell proliferation rate after THGP treatment decreased. What would be the reason? Could it be the toxicity of THGP? The authors should also test THGP effect on other macrophage cell lines and normal cell lines, such as J774 and 293 T cells.

2.  In Figure 2a, there was another paper showed that the THGP could decrease the IL-1β secretion, which is opposite of the authors  conclusion in this manuscript. Could the authors explain the reason? (https://www.ncbi.nlm.nih.gov/pmc/articles/PMC9654755/)

3.  In Figure 3j and 3k, the authors should include inhibitor JSH as a control group to prove that inhibition of NF-κB could directly decrease macrophages polarization to M1.

4. In Figure 5a and 5b, the authors should include B16 treatment with THGP as a control group to show that the single treatment didn't change the B16 cell number.

5. In Figure 6, what would be the mechanism of THGP decrease the expression of SIRP-α? Is it also through NF-κB pathway?

6. In Figure 7, the authors showed that the Ge-132 diet could promote the macrophage polarization to M1 in vivo. Did the authors test the effect of Ge-132 diet on other immune cells, such as CD8 T cells, CD4 cells and NK cells?

Author Response

To Reviewer 3

Thank you for your kind comments. I believe the manuscript was improved by the modifications. The responses to the comments are provided below.

  1.  In Figure 1d, the authors showed that the cell proliferation rate after THGP treatment decreased. What would be the reason? Could it be the toxicity of THGP? The authors should also test THGP effect on other macrophage cell lines and normal cell lines, such as J774 and 293 T cells.

A) In Figure 1d, RAW 264.7 cells were cultured with THGP-containing medium during induction of differentiation, but THGP-free medium was used during the proliferation assay. Therefore, we believe that the reason for the decreased proliferation is the differentiation into M1 macrophages of RAW 264.7 cells and not the cytotoxicity of THGP. Indeed, another paper reported that proliferation is reduced when RAW 264.7 cells are differentiated into M1 macrophages by LPS (Medical Gas Res8, 154, 2018). The above explanation has been added to the Discussion (Lines 370-376) and Materials and Methods (Lines 553-556).

Since we do not have a J774 cell line or other macrophage cell line, we cannot confirm the cytotoxicity of THGP. However, in unpublished data, THGP does not affect the proliferation of THP-1 cells (human monocyte cell line) up to 5 mM. In addition, we confirmed that THGP does not exhibit cytotoxicity up to 5 mM in normal human melanocytes, keratinocytes and fibroblasts.

  1.  In Figure 2a, there was another paper showed that the THGP could decrease the IL-1βsecretion, which is opposite of the authors conclusion in this manuscript. Could the authors explain the reason? (https://www.ncbi.nlm.nih.gov/pmc/articles/PMC9654755/)

A) Indeed, we reported that THGP decreased the enhanced IL-1β secretion by LPS and ATP. This mechanism is due to THGP complexed with ATP. However, in this study, long-term treatment with THGP induced macrophages to differentiate into M1 macrophages by promoting the nuclear translocation of NF-κB and increasing IL-1β expression. Therefore, we believe that THGP exhibits anti-inflammatory or immunostimulatory (so-called immune regulatory) effects through different mechanisms. We described this in the discussion (Lines 396-398).

  1. In Figure 3j and 3k, the authors should include inhibitor JSH as a control group to prove that inhibition of NF-κB could directly decrease macrophages polarization to M1.

A) As noted, an experiment was conducted with the addition of the JSH-only group as a control, and the figures were replaced.

  1. In Figure 5a and 5b, the authors should include B16 treatment with THGP as a control group to show that the single treatment didn't change the B16 cell number.

A) Thank you for your comments. In Figures 5a and 5b, all the groups were assayed in medium that did not contain THGP, so a THGP-only group was not considered necessary as a control. In addition, as shown in Figure 5g, THGP did not appear to be cytotoxic to B16 melanoma cells. The above explanation has been added to the Materials and Methods (Lines 664 and 667).

  1. In Figure 6, what would be the mechanism of THGP decrease the expression of SIRP-α? Is it also through NF-κB pathway?

A) We do not know why THGP reduced the expression of SIRP-α, so we would like to further investigate this in the future. We described this in the discussion (Lines 531-532).

  1. In Figure 7, the authors showed that the Ge-132 diet could promote the macrophage polarization to M1 in vivo. Did the authors test the effect of Ge-132 diet on other immune cells, such as CD8 T cells, CD4 cells and NK cells?

 A) We are also very interested in the effects of long-term administration of Ge-132 on other immune cells, such as NK cells and T cells. I would like to as address this issue in a future study. Ge-132 was also found to activate NK cells and macrophages in vivo after short-term exposures (24-72 h) (Microbiol Immunol. 1985;29(1):65–74.).

This manuscript is a resubmission of an earlier submission. The following is a list of the peer review reports and author responses from that submission.

Round 1

Reviewer 1 Report

This Manuscript aims at identifying the mechanism by which THGP promotes the differentiation of macrophages into M1 type and as a consequence also exerts enhanced cytotoxic activity against melanoma cells in vitro.

In my opinion, several points need to be addressed before considering the manuscript for publication.

1)      The Flow cytometry analysis performed to assess the size and complexity of macrophages after long exposure to THGP does not indicate what the AAs claim, as the physical parameters of the two populations (CTR vs THGP) look almost the same. Moreover, I’m also not convinced about the procedure of sample preparation. Why did the AAs fix the cells? This procedure usually slightly affects the cell size. I would suggest AAs to perform this assay with non-fixed cells.   

2)      In order to substantiate the effect of THGP in reverting macrophages phenotype after being exposed to LPS or IL-4, AAs need to show the quantification of CD86 and CD206.

3)      How the AAs assessed that the treatment with THGP determines an enhancement of macrophage adhesion? The graph in Figure 4b represents the % of adherent cells. In the M&M section AAs state that the adherent cells were determined through the MTS assay, which evaluates the viability of the cells. With respect to what is it to be intended the percentage of adherent cells?

Did the AAs have an idea of the mechanisms determining the increased adhesion of RAW 264.7?

4)       I have several concerns about the experiments whose results have been described in paragraph 2.5 “THGP Increases Cytotoxicity against B16 4A5 Mouse Melanoma Cells in RAW 264.7 Cells”.

I’m not convinced how the experiments in Figure 5a have been performed. If the co-culture of B16 4A5 and RAW 264.7 cells has been performed in the presence of THGP, and I think this is the case, the experimental condition represented by B16 4A5 plus THGP should be added. Further, the information on how AAs calculated the viability of B16 4A5 cells in co-culture with RAW 264.7 is missing. Indeed, Figure S3, to which the AAs refer for the above explanation, shows the expression of M1/M2 maker in macrophages from mice fed Ge-132. In my opinion, a model system employing for example luc-transfected cells, whose viability can be easily recorded by the addition of luciferin, could help to obtain more reliable results.  

Moreover, if I correctly understand the experiment shown in Figure 5f, AAs intended to determine also by flow cytometry the phagocytic activity of RAW 264.7 treated with THGP and then co-cultured with B16 4A5. I would expect to find an increase in the double positive cells (RAW 264.7 with phagocytosed B16 4A5). Is it a compensation problem? Why should the percentage of RAW 264.7 augment?

5)      The main conclusion of the study is that stimulation of macrophages with THGP leads to M1 polarization with a consequent increased phagocytic and cytotoxic activity of polarized macrophages against cancer cells. If the increased phagocytic activity has been demonstrated and the results obtained can support the conclusion drawn, I was wondering what the mechanism of cytotoxicity is, since the AAs concluded that the mechanism of cancer cell killing is not dependent on the release of soluble factors.

6)      Finally, as a minor comment, I think that the introduction section, especially the first part dedicated to the description of macrophages, their plasticity, and their differential role in inflammatory diseases and cancer, deserves the addition of references from AAs such as Pollard JW, Sica A, Mantovani A, and Gordon S which made the history in this very interesting field.  

Author Response

Thank you for your kind comments. I think the manuscript was improved by the modifications. I would like to respond as much as possible, but we experienced some difficulties in conducting additional tests. The responses to the comments are provided below.

  • The Flow cytometry analysis performed to assess the size and complexity of macrophages after long exposure to THGP does not indicate what the AAs claim, as the physical parameters of the two populations (CTR vs THGP) look almost the same. Moreover, I’m also not convinced about the procedure of sample preparation. Why did the AAs fix the cells? This procedure usually slightly affects the cell size. I would suggest AAs to perform this assay with non-fixed cells.
  1. Thank you for your suitable advice. Although the distribution of Ctrl and THGP cells was very similar, we proposed that some of the THGP-treated cells have a distribution that does not exist in Ctrl cells. Next, we needed a long time to prepare single-cell suspensions from a large number of cell cultures and to analyze them. Although the cell suspensions were maintained on ice, we worry about the effect of waiting time on the cell size. Therefore, we avoided the change in cell size by fixing the cells. We added the phrase "To avoid a change in cell size" to the M&M to avoid a misunderstanding. On the other hand, in the analysis using the Countess® II FL auto cell counter, live cells were analyzed because the analysis was completed easily.
  • In order to substantiate the effect of THGP in reverting macrophages phenotype after being exposed to LPS or IL-4, AAs need to show the quantification of CD86 and CD206.
  1. A) We quantified the images of immunofluorescence staining shown in Fig. 2. As a result, the cells treated with LPS exhibited a 7-fold increase in the expression of CD86, whereas LPS+THGP increased the expression of CD86 approximately 3-fold. On the other hand, when treated with IL-4, both IL-4 and IL-4+THGP approximately doubled the expression, and no difference was observed. These results have been added to Figure 2.

3)      How the AAs assessed that the treatment with THGP determines an enhancement of macrophage adhesion? The graph in Figure 4b represents the % of adherent cells. In the M&M section AAs state that the adherent cells were determined through the MTS assay, which evaluates the viability of the cells. With respect to what is it to be intended the percentage of adherent cells?

  1. A) RAW 264.7 cells were incubated for 30 minutes after Nonadherent cells were removed by washing twice with PBS (-), and the number of adherent cells was evaluated using the MTS assay. On the other hand, the total number of cells was evaluated using MTS assay without washing with PBS (-), and the total cells, including floating and adherent cells, were evaluated as 100%. I added this information to Section 4.12 in the M&M.

Did the AAs have an idea of the mechanisms determining the increased adhesion of RAW 264.7?

  1. A) M1 macrophages exhibit improved cell adhesion and the Myosin protein contributes to adhesion. Based on the results from our microarray analysis, the expression of Myosin 1D and Myosin F was 26.8- or 18.8-fold higher, respectively, in THGP-treated cells than in Ctrl cells (Sup. table 2). Therefore, the increase in Myosin protein expression increased adhesion.

4)       I have several concerns about the experiments whose results have been described in paragraph 2.5 “THGP Increases Cytotoxicity against B16 4A5 Mouse Melanoma Cells in RAW 264.7 Cells”.

I’m not convinced how the experiments in Figure 5a have been performed. If the co-culture of B16 4A5 and RAW 264.7 cells has been performed in the presence of THGP, and I think this is the case, the experimental condition represented by B16 4A5 plus THGP should be added. Further, the information on how AAs calculated the viability of B16 4A5 cells in co-culture with RAW 264.7 is missing. Indeed, Figure S3, to which the AAs refer for the above explanation, shows the expression of M1/M2 maker in macrophages from mice fed Ge-132.

  1. A) When B16 and RAW 264.7 cells were cocultured, they were cultured in THGP-free medium to eliminate the effect of THGP. Therefore, the results of the MTS assay of B16 4A5 cells cultured alone in Fig. 5a do not include the results of THGP treatment. The notation S3 listed in the M&M was a mistake. The results shown in Fig. 5a illustrate the absorbance of RAW 264.7 cells or B16 4A5 cultured alone or cocultured, as determined using MTS assay. The M&M was corrected to indicate the exact method.

In my opinion, a model system employing for example luc-transfected cells, whose viability can be easily recorded by the addition of luciferin, could help to obtain more reliable results.

  1. Thank you for presenting the experimental method in Fig. 5a. Unfortunately, our lab is not equipped to perform luciferase assays because we are not equipped to construct expression vectors. We will consider analyzing this issue in the future.

Moreover, if I correctly understand the experiment shown in Figure 5f, AAs intended to determine also by flow cytometry the phagocytic activity of RAW 264.7 treated with THGP and then co-cultured with B16 4A5. I would expect to find an increase in the double positive cells (RAW 264.7 with phagocytosed B16 4A5). Is it a compensation problem? Why should the percentage of RAW 264.7 augment?

  1. A) We used unstained cells as a negative control for gating. Therefore, we propose that the results presented in Fig. 5f-h show the correct values. In addition, we conducted this experiment multiple times and obtained similar results.

5)      The main conclusion of the study is that stimulation of macrophages with THGP leads to M1 polarization with a consequent increased phagocytic and cytotoxic activity of polarized macrophages against cancer cells. If the increased phagocytic activity has been demonstrated and the results obtained can support the conclusion drawn, I was wondering what the mechanism of cytotoxicity is, since the AAs concluded that the mechanism of cancer cell killing is not dependent on the release of soluble factors.

  1. A) As shown in Fig. 5i, we postulate that the increase in cytotoxicity is independent of secreted factors. As shown in Fig. 6a, THGP-treated macrophages appear to attack cancer cells directly and enhance the recognition of cancer cells. Therefore, macrophages may not only phagocytose cancer cells but also induce cytotoxicity by contacting cancer cells. We will continue to study the cytotoxicity induced by THGP.

6)      Finally, as a minor comment, I think that the introduction section, especially the first part dedicated to the description of macrophages, their plasticity, and their differential role in inflammatory diseases and cancer, deserves the addition of references from AAs such as Pollard JW, Sica A, Mantovani A, and Gordon S which made the history in this very interesting field.

  1. A) Thank you for noting this issue. We have added the paper you referenced to the Introduction.

Reviewer 2 Report

Authors have tried to address my previous considerations.

Minor comments.

- Figure 1a and 1d legends might be improved, they are not well explained.

- TAMs promote cancer metastasis by secreting TGF-β and increase angiogenesis by secreting vascular endothelial growth factor (VEGF). Therefore, the suppression of TAMs is an important target in cancer therapy [13]. These important items have been recently addressed in human melanoma and they do not occurr exactly as authors describe, reference to Gutiérrez-Seijo et al. JID 2021 should be included at line 59.

Author Response

Thank you for your kind comments. I think the manuscript was improved with the modifications.

The responses to the comments are provided below.

- Figure 1a and 1d legends might be improved, they are not well explained.

  1. A) Thank you for your advice. The legends of Fig. 1a, d are described in more detail.

- TAMs promote cancer metastasis by secreting TGF-β and increase angiogenesis by secreting vascular endothelial growth factor (VEGF). Therefore, the suppression of TAMs is an important target in cancer therapy [13]. These important items have been recently addressed in human melanoma and they do not occurr exactly as authors describe, reference to Gutiérrez-Seijo et al. JID 2021 should be included at line 59.

  1. A) Thank you for your advice. The paper titled “Activin A Sustains the Metastatic Phenotype of Tumor-Associated Macrophages and Is a Prognostic Marker in Human Cutaneous Melanoma” was added to the citations.

Round 2

Reviewer 1 Report

Below are a few more comments to the AAs reply.

Thank you for your kind comments. I think the manuscript was improved by the modifications. I would like to respond as much as possible, but we experienced some difficulties in conducting additional tests. The responses to the comments are provided below.

1)        The Flow cytometry analysis performed to assess the size and complexity of macrophages after long exposure to THGP does not indicate what the AAs claim, as the physical parameters of the two populations (CTR vs THGP) look almost the same. Moreover, I’m also not convinced about the procedure of sample preparation. Why did the AAs fix the cells? This procedure usually slightly affects the cell size. I would suggest AAs to perform this assay with non-fixed cells.

A)    Thank you for your suitable advice. Although the distribution of Ctrl and THGP cells was very similar, we proposed that some of the THGP-treated cells have a distribution that does not exist in Ctrl cells. Next, we needed a long time to prepare single-cell suspensions from a large number of cell cultures and to analyze them. Although the cell suspensions were maintained on ice, we worry about the effect of waiting time on the cell size. Therefore, we avoided the change in cell size by fixing the cells. We added the phrase "To avoid a change in cell size" to the M&M to avoid a misunderstanding. On the other hand, in the analysis using the Countess® II FL auto cell counter, live cells were analyzed because the analysis was completed easily.

The distribution of the cell population, in the flow cytometry dot plot, indicates only that the two populations are overlapping in size, with a slight difference in their complexity.

2)        In order to substantiate the effect of THGP in reverting macrophages phenotype after being exposed to LPS or IL-4, AAs need to show the quantification of CD86 and CD206.

A) We quantified the images of immunofluorescence staining shown in Fig. 2. As a result, the cells treated with LPS exhibited a 7-fold increase in the expression of CD86, whereas LPS+THGP increased the expression of CD86 approximately 3-fold. On the other hand, when treated with IL-4, both IL-4 and IL-4+THGP approximately doubled the expression, and no difference was observed. These results have been added to Figure 2.

3)      How the AAs assessed that the treatment with THGP determines an enhancement of macrophage adhesion? The graph in Figure 4b represents the % of adherent cells. In the M&M section AAs state that the adherent cells were determined through the MTS assay, which evaluates the viability of the cells. With respect to what is it to be intended the percentage of adherent cells?

A) RAW 264.7 cells were incubated for 30 minutes after seeding. Nonadherent cells were removed by washing twice with PBS (-), and the number of adherent cells was evaluated using the MTS assay. On the other hand, the total number of cells was evaluated using MTS assay without washing with PBS (-), and the total cells, including floating and adherent cells, were evaluated as 100%. I added this information to Section 4.12 in the M&M.

I thank the AAs for explaining to me how they performed the experiments. Nevertheless, I think that the determination of the percentage of adherent cells in each experimental group should be determined by counting the total number of cells per well (floating  plus adherent).

Did the AAs have an idea of the mechanisms determining the increased adhesion of RAW 264.7?

A)    M1 macrophages exhibit improved cell adhesion and the Myosin protein contributes to adhesion. Based on the results from our microarray analysis, the expression of Myosin 1D and Myosin F was 26.8- or 18.8-fold higher, respectively, in THGP-treated cells than in Ctrl cells (Sup. table 2). Therefore, the increase in Myosin protein expression increased adhesion.

The increased expression of myosin is relative to RNA and not protein expression. The AAs cannot say “Therefore, the increase in Myosin protein expression increased adhesion” because this, although reasonable, is only a speculation and it should be reported as such. If the AAs want to claim that an increased expression of Myosin is responsible for enhanced adhesion, they should investigate what happens by silencing it. 

4)       I have several concerns about the experiments whose results have been described in paragraph 2.5 “THGP Increases Cytotoxicity against B16 4A5 Mouse Melanoma Cells in RAW 264.7 Cells”.

I’m not convinced how the experiments in Figure 5a have been performed. If the co-culture of B16 4A5 and RAW 264.7 cells has been performed in the presence of THGP, and I think this is the case, the experimental condition represented by B16 4A5 plus THGP should be added. Further, the information on how AAs calculated the viability of B16 4A5 cells in co-culture with RAW 264.7 is missing. Indeed, Figure S3, to which the AAs refer for the above explanation, shows the expression of M1/M2 maker in macrophages from mice fed Ge-132.

A) When B16 and RAW 264.7 cells were cocultured, they were cultured in THGP-free medium to eliminate the effect of THGP. Therefore, the results of the MTS assay of B16 4A5 cells cultured alone in Fig. 5a do not include the results of THGP treatment. The notation S3 listed in the M&M was a mistake. The results shown in Fig. 5a illustrate the absorbance of RAW 264.7 cells or B16 4A5 cultured alone or cocultured, as determined using MTS assay. The M&M was corrected to indicate the exact method.

In my opinion, a model system employing for example luc-transfected cells, whose viability can be easily recorded by the addition of luciferin, could help to obtain more reliable results.

The system used by the AAs is not an appropriate assay to determine cytotoxicity. I can understand the point of the AAs, but they should find another way for the correct determination of what they are looking for.

A)    Thank you for presenting the experimental method in Fig. 5a. Unfortunately, our lab is not equipped to perform luciferase assays because we are not equipped to construct expression vectors. We will consider analyzing this issue in the future.

Moreover, if I correctly understand the experiment shown in Figure 5f, AAs intended to determine also by flow cytometry the phagocytic activity of RAW 264.7 treated with THGP and then co-cultured with B16 4A5. I would expect to find an increase in the double positive cells (RAW 264.7 with phagocytosed B16 4A5). Is it a compensation problem? Why should the percentage of RAW 264.7 augment?

A) We used unstained cells as a negative control for gating. Therefore, we propose that the results presented in Fig. 5f-h show the correct values. In addition, we conducted this experiment multiple times and obtained similar results.

As THGP does not affect the growth of RAW 264.7, as demonstrated in the MTS assay (Fig 5a) and underlined by the AAs themselves (“No difference in cell growth 231 was observed between RAW 264.7 cells cultured with or without THGP”) at line 231, how is it possible that the % of RAW 264.7 treated with THGP augments when co-cultured with B16 4A5? These two conclusions disagree with each other. Could it be due to the fact that B16 induces the proliferation of RAW?

5)      The main conclusion of the study is that stimulation of macrophages with THGP leads to M1 polarization with a consequent increased phagocytic and cytotoxic activity of polarized macrophages against cancer cells. If the increased phagocytic activity has been demonstrated and the results obtained can support the conclusion drawn, I was wondering what the mechanism of cytotoxicity is, since the AAs concluded that the mechanism of cancer cell killing is not dependent on the release of soluble factors.

 A) As shown in Fig. 5i, we postulate that the increase in cytotoxicity is independent of secreted factors. As shown in Fig. 6a, THGP-treated macrophages appear to attack cancer cells directly and enhance the recognition of cancer cells. Therefore, macrophages may not only phagocytose cancer cells but also induce cytotoxicity by contacting cancer cells. We will continue to study the cytotoxicity induced by THGP.

6)      Finally, as a minor comment, I think that the introduction section, especially the first part dedicated to the description of macrophages, their plasticity, and their differential role in inflammatory diseases and cancer, deserves the addition of references from AAs such as Pollard JW, Sica A, Mantovani A, and Gordon S which made the history in this very interesting field.

A)  Thank you for noting this issue. We have added the paper you referenced to the Introduction.

Author Response

Thank you for pointing it out again. Responses to our comments are described in purple.

Thank you for your kind comments. I think the manuscript was improved by the modifications. I would like to respond as much as possible, but we experienced some difficulties in conducting additional tests. The responses to the comments are provided below.

1)        The Flow cytometry analysis performed to assess the size and complexity of macrophages after long exposure to THGP does not indicate what the AAs claim, as the physical parameters of the two populations (CTR vs THGP) look almost the same. Moreover, I’m also not convinced about the procedure of sample preparation. Why did the AAs fix the cells? This procedure usually slightly affects the cell size. I would suggest AAs to perform this assay with non-fixed cells.

  1. A)Thank you for your suitable advice. Although the distribution of Ctrl and THGP cells was very similar, we proposed that some of the THGP-treated cells have a distribution that does not exist in Ctrl cells. Next, we needed a long time to prepare single-cell suspensions from a large number of cell cultures and to analyze them. Although the cell suspensions were maintained on ice, we worry about the effect of waiting time on the cell size. Therefore, we avoided the change in cell size by fixing the cells. We added the phrase "To avoid a change in cell size" to the M&M to avoid a misunderstanding. On the other hand, in the analysis using the Countess® II FL auto cell counter, live cells were analyzed because the analysis was completed easily.

The distribution of the cell population, in the flow cytometry dot plot, indicates only that the two populations are overlapping in size, with a slight difference in their complexity.

Thank you for your advice. In Result 2.1, I rewrote that there was a slight difference in complexity from the results of flowcytometry under your suggestion. (Line 96-100)

2)        In order to substantiate the effect of THGP in reverting macrophages phenotype after being exposed to LPS or IL-4, AAs need to show the quantification of CD86 and CD206.

  1. A) We quantified the images of immunofluorescence staining shown in Fig. 2. As a result, the cells treated with LPS exhibited a 7-fold increase in the expression of CD86, whereas LPS+THGP increased the expression of CD86 approximately 3-fold. On the other hand, when treated with IL-4, both IL-4 and IL-4+THGP approximately doubled the expression, and no difference was observed. These results have been added to Figure 2.

3)      How the AAs assessed that the treatment with THGP determines an enhancement of macrophage adhesion? The graph in Figure 4b represents the % of adherent cells. In the M&M section AAs state that the adherent cells were determined through the MTS assay, which evaluates the viability of the cells. With respect to what is it to be intended the percentage of adherent cells?

  1. A) RAW 264.7 cells were incubated for 30 minutes after seeding. Nonadherent cells were removed by washing twice with PBS (-), and the number of adherent cells was evaluated using the MTS assay. On the other hand, the total number of cells was evaluated using MTS assay without washing with PBS (-), and the total cells, including floating and adherent cells, were evaluated as 100%. I added this information to Section 4.12 in the M&M.

I thank the AAs for explaining to me how they performed the experiments. Nevertheless, I think that the determination of the percentage of adherent cells in each experimental group should be determined by counting the total number of cells per well (floating  plus adherent).

I apologize for the lack of M&M explanations. As you pointed out, I rewrote as "The number of cells was evaluated using the MTS assay. The determination of the percentage of adherent cells in each experimental group was determined by counting the total number of cells per well (floating plus adherent cells)." (Line 621-624)

Did the AAs have an idea of the mechanisms determining the increased adhesion of RAW 264.7?

  1. A)    M1 macrophages exhibit improved cell adhesion and the Myosin protein contributes to adhesion. Based on the results from our microarray analysis, the expression of Myosin 1D and Myosin F was 26.8- or 18.8-fold higher, respectively, in THGP-treated cells than in Ctrl cells (Sup. table 2). Therefore, the increase in Myosin protein expression increased adhesion.

The increased expression of myosin is relative to RNA and not protein expression. The AAs cannot say “Therefore, the increase in Myosin protein expression increased adhesion” because this, although reasonable, is only a speculation and it should be reported as such. If the AAs want to claim that an increased expression of Myosin is responsible for enhanced adhesion, they should investigate what happens by silencing it. 

Thank you for pointing this out. Since the results of this experiment were predicted from the microarray results, we have reworded to "The increase of myosin genes expression may have enhanced the adhesion of THGP-treated cells.” (Line 421-422)

4)       I have several concerns about the experiments whose results have been described in paragraph 2.5 “THGP Increases Cytotoxicity against B16 4A5 Mouse Melanoma Cells in RAW 264.7 Cells”.

I’m not convinced how the experiments in Figure 5a have been performed. If the co-culture of B16 4A5 and RAW 264.7 cells has been performed in the presence of THGP, and I think this is the case, the experimental condition represented by B16 4A5 plus THGP should be added. Further, the information on how AAs calculated the viability of B16 4A5 cells in co-culture with RAW 264.7 is missing. Indeed, Figure S3, to which the AAs refer for the above explanation, shows the expression of M1/M2 maker in macrophages from mice fed Ge-132.

  1. A) When B16 and RAW 264.7 cells were cocultured, they were cultured in THGP-free medium to eliminate the effect of THGP. Therefore, the results of the MTS assay of B16 4A5 cells cultured alone in Fig. 5a do not include the results of THGP treatment. The notation S3 listed in the M&M was a mistake. The results shown in Fig. 5a illustrate the absorbance of RAW 264.7 cells or B16 4A5 cultured alone or cocultured, as determined using MTS assay. The M&M was corrected to indicate the exact method.

In my opinion, a model system employing for example luc-transfected cells, whose viability can be easily recorded by the addition of luciferin, could help to obtain more reliable results.

The system used by the AAs is not an appropriate assay to determine cytotoxicity. I can understand the point of the AAs, but they should find another way for the correct determination of what they are looking for.

Thank you for your advice. As it is not clear from the result of Fig.5a which RAW 264.7 or B16 4A5 cells decrease, the expression of cytotoxicity” invites misunderstanding. Therefore, "The results revealed that THGP-treated RAW 264.7 cells significantly suppressed cancer cell growth" was changed to "The total cell number of co-cultured with B16 4A5 cells was lower in THGP-treated cells than in Ctrl cells." (Line 234-235)

  1. A)    Thank you for presenting the experimental method in Fig. 5a. Unfortunately, our lab is not equipped to perform luciferase assays because we are not equipped to construct expression vectors. We will consider analyzing this issue in the future.

Moreover, if I correctly understand the experiment shown in Figure 5f, AAs intended to determine also by flow cytometry the phagocytic activity of RAW 264.7 treated with THGP and then co-cultured with B16 4A5. I would expect to find an increase in the double positive cells (RAW 264.7 with phagocytosed B16 4A5). Is it a compensation problem? Why should the percentage of RAW 264.7 augment?

  1. A) We used unstained cells as a negative control for gating. Therefore, we propose that the results presented in Fig. 5f-h show the correct values. In addition, we conducted this experiment multiple times and obtained similar results.

As THGP does not affect the growth of RAW 264.7, as demonstrated in the MTS assay (Fig 5a) and underlined by the AAs themselves (“No difference in cell growth 231 was observed between RAW 264.7 cells cultured with or without THGP”) at line 231, how is it possible that the % of RAW 264.7 treated with THGP augments when co-cultured with B16 4A5? These two conclusions disagree with each other. Could it be due to the fact that B16 induces the proliferation of RAW?

We think that the expression on FIgure5a confuses the issue. Although the absorbance in the RAW cells co-cultured with B16 cells increase, the absorbance in the co-culture shows the addition of B16 absorbance to the absorbance of RAW cells. Therefore, we think that B16 don't induce the proliferation of RAW. In contrast, the absorbance in the co-culture of RAW treated with THGP and B16 cells is lower than the additional absorbance of B16 and RAW cells, and may be equal to the absorbance of RAW treated with THGP. Therefore, we think that B16 cells might not grow. However, it is not clear which RAW 264.7 or B16 4A5 cells decrease. As mentioned above, we added "The total cell number of co-cultured with B16 4A5 cells was lower in THGP-treated cells than in Ctrl cells." in Line 234-235. We also added “On the other hand, the percentage of cells with RAW 264.7 cells increased. This result may be due to the relative increase in the percentage of RAW 264.7 cells because the number of B16 4A5 cells decreased (Figure 5h)." in Line 246-252).

5)      The main conclusion of the study is that stimulation of macrophages with THGP leads to M1 polarization with a consequent increased phagocytic and cytotoxic activity of polarized macrophages against cancer cells. If the increased phagocytic activity has been demonstrated and the results obtained can support the conclusion drawn, I was wondering what the mechanism of cytotoxicity is, since the AAs concluded that the mechanism of cancer cell killing is not dependent on the release of soluble factors.

  1. A) As shown in Fig. 5i, we postulate that the increase in cytotoxicity is independent of secreted factors. As shown in Fig. 6a, THGP-treated macrophages appear to attack cancer cells directly and enhance the recognition of cancer cells. Therefore, macrophages may not only phagocytose cancer cells but also induce cytotoxicity by contacting cancer cells. We will continue to study the cytotoxicity induced by THGP.

6)      Finally, as a minor comment, I think that the introduction section, especially the first part dedicated to the description of macrophages, their plasticity, and their differential role in inflammatory diseases and cancer, deserves the addition of references from AAs such as Pollard JW, Sica A, Mantovani A, and Gordon S which made the history in this very interesting field.

  1. A)  Thank you for noting this issue. We have added the paper you referenced to the Introduction.
